

# Constraining elemental mercury air–sea exchange using long-term ground-based observations

Koketso M. Molepo[1], Johannes Bieser[2], Alkuin M. Koenig[2,4], Ian M. Hedgecock[3], Ralf Ebinghaus[1], Aurélien Dommergue[4], Olivier Magand[5], Hélène Angot[4], Oleg Travnikov[6], Lynwill Martin[7,9], Casper Labuschagne[7], Katie Read[8], Yann Bertrand[4]

[1]Institute for Coastal Environmental Chemistry, Organic Environmental Chemistry, Helmholtz-Zentrum Hereon, Max-Planck-Str. 1, 21502 Geesthacht, Germany
[2]Institute of Coastal Systems–Analysis and Modeling, Helmholtz-Zentrum Hereon, Max-Planck-Str. 1, 21502 Geesthacht, Germany
[3]CNR-Institute of Atmospheric Pollution Research, 87036 Rende, Italy
[4]Univ. Grenoble Alpes, CNRS, INRAE, IRD, Grenoble INP*, IGE, Grenoble, France (*Institute of Engineering and Management Univ. Grenoble Alpes)
[5]Observatoire des Sciences de l'Univers de La Réunion (OSU-Réunion), UAR 3365, Université de La Réunion, CNRS, Météo France, IRD, Saint-Denis, France
[6]Department of Environmental Sciences, Jožef Stefan Institute, Jamova cesta 39, 1000 Ljubljana, Slovenia
[7]Cape Point Global Atmosphere Watch, South African Weather Service, Oval Park Office, Freight Road, Cape Town, 7525
[8]National Centre for Atmospheric Science, University of York, York, YO10 5DD, United Kingdom
[9]Atmospheric Chemistry Research Group, Chemical Resource Beneficiation, North-West University, Potchefstroom, South Africa

*Correspondence to*: Koketso M. Molepo (koketso.molepo56@gmail.com)

**Abstract.** Air–sea exchange of gaseous elemental mercury ($Hg^0$) is a major component of the global mercury (Hg) biogeochemical cycle but remains poorly understood due to sparse in situ measurements. Here, we used long-term atmospheric $Hg^0$ ($Hg^0_{air}$) observations combined with air mass back trajectories at four ground-based monitoring sites to study $Hg^0$ air–sea exchange. The trajectories showed that all four sites sample mainly marine air masses. At all sites, we observed a gradual increase in mean $Hg^0_{air}$ concentration with air mass recent residence time in the Marine Boundary Layer (MBL), followed by a steady state. The pattern is consistent with the thin film gas exchange model, which predicts net $Hg^0$ emissions from the surface ocean until the $Hg^0_{air}$ concentration normalised by Henry's law constant matches the surface ocean dissolved $Hg^0$ ($Hg^0_{aq}$) concentration. This provides strong evidence that ocean $Hg^0$ emissions directly influence $Hg^0_{air}$ concentrations at these sites. Using the observed relationship between $Hg^0_{air}$ concentrations and air mass recent MBL residence time, we estimated mean surface ocean $Hg^0_{aq}$ concentrations of ~ 4–7 pg L$^{-1}$ for the North Atlantic and Arctic oceans (AA) and ~ 4 pg L$^{-1}$ for the Southern, South Atlantic and south Indian oceans (SSI). Estimated ocean $Hg^0$ emission fluxes ranged between 0.58–0.75 and 0.47–0.66 ng m$^{-2}$ h$^{-1}$ for the AA and SSI, respectively, with a global extrapolated mean flux of around 1900 t y$^{-1}$ (1200–2600 t y$^{-1}$). This study demonstrates the applicability of long-term, ground-based $Hg^0_{air}$ observations in constraining $Hg^0$ air–sea exchange.



## 1 Introduction

Mercury (Hg), a ubiquitous element in the environment, has gained widespread attention due to its adverse effects on human health and ecosystems (AMAP/UNEP, 2018; Budnik and Castelyn, 2019; Al-Sulaiti et al., 2022; Basu et al., 2023). While naturally occurring in the Earth's crust and released by natural processes, anthropogenic activities have dramatically increased the amount of Hg emitted into the environment (Sunderland and Mason, 2007; Amos et al., 2013; Lamborg et al., 2014; Zhang et al., 2014; Streets et al., 2017; Li et al., 2020; Sonke et al., 2023). Consequently, Hg has become a persistent global pollutant.

The elemental form of Hg ($Hg^0$) is particularly important. Owing to its chemical inertness and low water solubility, once emitted, $Hg^0$ stays in the atmosphere for several months to a year (Horowitz et al., 2017; Saiz-Lopez et al., 2018; Shah et al., 2021), allowing it to travel thousands of kilometres across the globe, impacting even remote areas (Travnikov 2005; Durnford et al., 2010; Koenig et al., 2022). Although in the atmosphere $Hg^0$ exists in low concentrations with no direct harmful effects (Holloway and Littlefield, 2011), a portion of the atmospheric $Hg^0$ is gradually oxidised to more soluble divalent Hg compounds ($Hg^{II}$) that are readily deposited to surface reservoirs (Horowitz et al., 2017; Saiz-Lopez et al., 2018; Shah et al., 2021). In aquatic environments, some of this deposited $Hg^{II}$ is converted to methylmercury (Lehnherr, 2014), a highly toxic compound that bioaccumulates and biomagnifies up the food chain (Lehnherr, 2014), posing serious health risks to humans through the consumption of seafood (Driscoll et al., 2013; Zillioux, 2015; AMAP/UNEP, 2018; Al-Sulaiti et al., 2022).

The exchange of $Hg^0$ between the atmosphere and the oceans plays a crucial role in the cycling of Hg in the environment. Atmospheric deposition, being the dominant source of Hg ($Hg^0$, as well as $Hg^{II}$) entering the ocean (Horowitz et al., 2017; AMAP/UNEP, 2018; Zhang et al., 2019, 2023; Jiskra et al., 2021; Shah et al., 2021), strongly influences the distribution of aqueous Hg (Strode et al., 2007; Kuss et al., 2011; Soerensen et al., 2014). On the other hand, most of the deposited Hg is eventually re-emitted back to the atmosphere, with ocean emissions constituting about one-third of the total annual Hg released to the atmosphere (Horowitz et al., 2017; AMAP/UNEP, 2018; Zhang et al., 2019; Shah et al., 2021). Through this "multihop" mechanism (Hedgecock and Pirrone, 2004), where atmospheric Hg is deposited to the ocean and the ocean re-emits it back to the atmosphere (as $Hg^0$), ocean emissions contribute to the long-range transport of $Hg^0$ and extend the lifetime of Hg actively cycling in the environment (Strode et al., 2007; Amos et al., 2013). Conversely, reduction of dissolved $Hg^{II}$ to $Hg^0$ in the water column and its subsequent evasion decreases the $Hg^{II}$ reservoir available for conversion to methylmercury (Lehnherr, 2014). Evidently, accurately characterising $Hg^0$ air–sea exchange is critical.

Our understanding of $Hg^0$ air–sea exchange and cycling in the Marine Boundary Layer (MBL) mainly derives from in situ measurements performed during ship-board sampling campaigns (e.g., Gårdfeldt et al., 2003; Andersson et al., 2007, 2008b, 2011; Kuss et al., 2011; Soerensen et al., 2013, 2014; Mason et al., 2017; Nerentorp Mastromonaco et al., 2017; Wang et al., 2017; Wang et al., 2019). These observations, comprising of simultaneous measurements of $Hg^0$ in air ($Hg^0_{air}$) and in surface



seawater ($Hg^0_{aq}$), along with variables such as sea surface temperature (SST) and wind speed, have been instrumental in discerning the spatio-temporal variability of $Hg^0$ at the air–sea interface, estimating $Hg^0$ air–sea exchange fluxes, as well as understanding the factors driving these processes. However, these measurements are sparse, covering only short periods (typically a few days to a few weeks), and are predominantly performed in the northern hemisphere. Chemical transport models can provide insights beyond the spatio-temporal limitations of in situ measurements, but they are also limited by the sparsity

in direct measurements, such as for validating air–sea exchange parameterisations applied in the models (e.g., Zhang et al., 2019). As a result, many aspects of $Hg^0$ dynamics at the air–sea interface remain poorly understood.

Some ground-based monitoring sites sample air masses with extended ocean exposure, offering a valuable opportunity to study $Hg^0$ cycling at the air–sea interface. Moreover, many of these sites have continuous $Hg^0_{air}$ records spanning several years (e.g.,

Sprovieri et al., 2016). Despite their potential, their application for extensive analyses of $Hg^0$ air–sea exchange remains underexplored, with studies primarily using them for model validation (e.g., Soerensen et al., 2010b) or for complementing ship-based measurements (e.g., Gårdfeldt et al., 2003; Sommar et al., 2010).

In this paper, we analyse $Hg^0_{air}$ observations from four ground-based monitoring sites to study $Hg^0$ air–sea exchange. We use

measurements from the longest continuous monitoring sites in the northern hemisphere (NH) and southern hemisphere (SH), Mace Head and Cape Point, respectively, as well as from the sub-tropical North Atlantic site Cabo Verde Observatory and the remote southern Indian Ocean site Amsterdam Island. The observations are combined with air mass back trajectories to gain insights into the sources and travel paths of air masses sampled at the sites. The results from this study will contribute towards an improved understanding of $Hg^0$ air–sea exchange.

**2 Data and methods**

**2.1 Observations**

**2.1.1 Site descriptions**

We used observations from four monitoring sites, namely, Mace Head, Cabo Verde Observatory, Cape Point and Amsterdam Island (Fig. 1). Below is a brief description of each site.


*Mace Head*
Mace Head (53°20′ N, 9°54′ W) is located in County Galway on the west coast of Ireland (Ebinghaus et al., 2011). It is exposed to the North Atlantic Ocean with a wide clean sector between 180° and 300° (Ebinghaus et al., 2011), ideally situated to study atmospheric composition under northern hemispheric background conditions but also under regionally polluted European

continental conditions, when air masses originate from an easterly direction (Ebinghaus et al., 2011). On average, over 50 %



of air masses that arrive at Mace Head are within the clean sector, having recently travelled thousands of kilometres across the North Atlantic Ocean (Ebinghaus et al., 2011). The climate at Mace Head is mild and moist (www.macehead.org, last access: June 2022), with an average air temperature of about 10 °C, generally high relative humidity (80–85 %) and an annual rainfall of approximately 1200 mm. Air sampling at the site is performed from two towers (10 m and 22 m high) located 100 m from

the shoreline and 50 m from the high-water mark. There are no industrial activities that could influence the measurements at Mace Head (Ebinghaus et al., 2011), with the nearest major urban area, Galway city, situated about 90 km east of the station (Ebinghaus et al., 2011). Mace Head is operated by the Atmospheric Science Research Group at the National University of Ireland, Galway (Kock et al., 2005), and is part of multiple international research networks, including the Global Mercury Observation System (GMOS) and the Global Atmosphere Watch (GAW) of the World Meteorological Organisation (WMO).


### *Cabo Verde Observatory*

Cabo Verde Observatory (16°51′ N, 24°52′ W) lies on the northeast side of São Vicente (Read et al., 2017), one of ten islands in the Cabo Verde archipelago in the North Atlantic Ocean (Read et al., 2017). The site is characterised by a warm and dry climate, with a mean annual air temperature of $24 \pm 2$ °C and annual rainfall below 200 mm, most of which occurs in the rainy

season of July–November (Read et al., 2017). The observatory receives mostly (about 95 % of the time) air masses from the northeasterly trade winds, which have typically travelled for five days over the ocean (Read et al., 2017), providing the opportunity to study clean marine air (Carpenter et al., 2010; Read et al., 2017). Air sampling at Cabo Verde Observatory is performed 50 m from the coastline at 10 m above sea level (Carpenter et al., 2010). The observatory is operated as a multilateral project between the United Kingdom, Germany and the Republic of Cabo Verde (Read et al., 2017), and is also part of both

the GMOS and WMO-GAW network of sites (Carpenter et al., 2010; Sprovieri et al., 2016; Read et al., 2017).

### *Cape Point*

Cape Point (34°21′ S, 18°29′ E) is located on the southern tip of the Cape Peninsula, about 60 km south of Cape Town, South Africa (Martin et al., 2017). It is situated within the Cape Point National Park, on a coastal cliff 230 m above sea level (Brunke

et al., 2004; Martin et al., 2017). The site has a Mediterranean-type climate, characterised by moderate temperatures, dry summers and increased precipitation during winter (Brunke et al., 2004). The prevailing wind direction at Cape Point is from the southeast to southwest, and thus most air masses advected to the site are clean marine air from the Southern Ocean (Brunke et al., 2004). On average, over 50 % of the air masses reaching Cape Point originate from this clean oceanic sector, having recently travelled thousands of kilometres across the southern oceanic regions (Labuschagne et al., 2018). Nevertheless, the

site occasionally experiences air masses from the north to northeast, mainly in winter, that are influenced by anthropogenic emissions from the greater Cape Town area and/or by other continental sources (Brunke et al., 2010). Cape Point is operated by the South African Weather Service (SAWS) and is also part of the GMOS and WMO-GAW networks (Martin et al., 2017; Slemr et al., 2020).





*Amsterdam Island*

Amsterdam Island (37°48′ S, 77°34′ E) is a small subtropical island (55 km$^2$) in the southern Indian Ocean, located about 3400 km and 5000 km downwind of Madagascar and South Africa, respectively (Angot et al., 2014; Slemr et al., 2015, 2020; Li et al., 2023; Magand et al., 2023; Tassone et al., 2023). The  atmospheric monitoring station is situated at Pointe Bénédicte at the northwest end of the island at an altitude of 70 m above sea level (Angot et al., 2014; Magand et al., 2023).  Air sampling and monitoring at the Pointe Bénédicte station is performed from different heights depending on the pollutant studied, with atmospheric Hg being sampled at 6 m above ground level. The site has an oceanic climate with mild temperatures (ranging between 11 and 17 °C over the year), high relative humidity (65 to 85 %) and frequent presence of clouds and abundant rainfall (total annual average between ~800 mm and ~1300 mm from the last decades) (Miller et al., 1993; Angot et al., 2014; Tassone et al., 2023). The dominant surface wind direction at Amsterdam Island is westerly and northwesterly, and wind speeds are comparatively high throughout the year (with an annual average of 7.6 m s$^{-1}$; Miller et al., 1993), peaking in austral winter (Baboukas et al., 2004; Li et al., 2023). The island is largely free of human disturbance. The only possible, but very rare, local source of pollution may emanate from the 20–40 scientific research and technical crew in Martin-de-Viviès life base 2 kilometres downwind of the Pointe Bénédicte station. Amsterdam Island is running under administration of Terres Australes et Antarctiques Françaises (TAAF), the French Southern and Antarctic Lands, and is scientifically operated by the French Polar Institute (IPEV) research programs, all year round. The monitoring station is part of the French national monitoring system, the Integrated Carbon Observation System, ICOS-France Atmosphère for the long-term observation of greenhouse gases (El Yazidi et al., 2018; Magand et al., 2023). As with Mace Head, Cabo Verde Observatory and Cape Point, the Pointe Bénédicte station in Amsterdam Island is part of both the GMOS network and is also labelled as a WMO-GAW site (Angot et al., 2014; Slemr et al., 2015, 2020; Magand et al., 2023).

**2.1.2 Atmospheric Hg measurements**

Atmospheric Hg has been measured since September 1995, December 2011, March 2007 and January 2012 at Mace Head, Cabo Verde Observatory, Cape Point and Amsterdam Island, respectively, using Tekran 2537 A/B model analysers (Tekran Inc., Toronto, Canada; Ebinghaus et al., 2002; Read et al., 2017; Slemr et al., 2020; Magand et al., 2023). The analysers are based on Hg enrichment on a gold cartridge, followed by thermal desorption and detection by cold vapour atomic fluorescence spectroscopy (CVAFS) at 253.7 nm (Fitzgerald and Gill, 1979; Bloom and Fitzgerald, 1988). The analysers use two gold cartridges, switching between the cartridges to allow for alternating sampling and desorption, resulting in continuous sampling of the incoming air (Ebinghaus et al., 2011; Martin et al. 2017; Slemr et al., 2020; Magand et al., 2023). Concentrations are expressed in nanograms per cubic metre at standard temperature and pressure (STP) conditions (273.15 K, 1013.25 hPa) with an instrumental detection limit close to 0.1 ng m$^{-3}$ and an Hg$^0_{air}$ average uncertainty value of about 10% (Slemr et al., 2015).

The instruments are automatically calibrated every 25 h at Mace Head and Cape Point, every 72 h at Cabo Verde Observatory and every 69 h at Amsterdam Island using internal permeation sources (Ebinghaus et al., 2011; Read et al., 2017; Slemr et al,



2020; Magand et al., 2023). The permeation sources are in turn checked (annually at Cabo Verde Observatory and Cape Point, bi-annually at Mace Head and quarterly at Amsterdam Island) by manual injections of saturated Hg vapour from a temperature-

controlled vessel adapted following Dumarey et al. (1985) procedures (Ebinghaus et al., 2011; Slemr et al, 2020; Magand et al., 2023). To ensure comparable results, the Tekran analysers at all four sites are operated and calibrated under GMOS standard operating procedures (SOPs; Sprovieri et al., 2016; Martin et al., 2017; Read et al., 2017; Slemr et al., 2020). The sampling procedure, quality assurance and quality control measures in operation at the sites are detailed in several references (e.g., Ebinghaus et al., 2011; Angot et al., 2014; Sprovieri et al., 2016; Slemr et al., 2020; Magand et al., 2023).


Several studies referenced here, including some of which use the same atmospheric Hg observations as those in the present study (i.e., from the same monitoring sites), denote (the) Tekran 2537A/B measurements as total gaseous mercury (TGM; $TGM = Hg^0_{air}$ + gaseous oxidised mercury, $Hg^{II}_{air}$) (e.g., Gårdfeldt et al., 2003; Temme et al., 2003; Kock et al., 2005; Xia et al., 2010; Andersson et al., 2011; Ebinghaus et al., 2011; Read et al., 2017). However, at all four sites studied here, it has been

suggested that the measurements are most likely of $Hg^0_{air}$ rather than TGM, as $Hg^{II}_{air}$ is easily lost to sea-salt deposits in the inlet tubing and filters of the instrument (Weigelt et al., 2015; Read et al., 2017; Slemr et al., 2020). In Amsterdam Island, between 2012 and 2015 atmospheric mercury species were measured using a Tekran Hg speciation unit (Tekran 1130/1135 models), ensuring that only $Hg^0_{air}$ is sampled and analysed with the associated Tekran 2537A/B model. Since the uninstallation of the speciation unit at the site in 2015, the Tekran 2537A/B instrument has been deployed with a specific setup using two

0.45 μm polyethersulfone cation-exchange membranes (PES-CEM, 0.45 μm, 47 mm, Merck Millipore®) installed at the inlet of the heated line, preventing the introduction of oxidised species and ensuring that only $Hg^0_{air}$ is measured (Magand et al., 2023). Furthermore, TGM typically comprises mostly (> 98 %) $Hg^0_{air}$ (e.g., Soerensen et al., 2010a; Cheng et al., 2014; Yin et al., 2018; Wang et al., 2019). Therefore, we regard the Tekran 2537A/B measurements at the four sites as $Hg^0_{air}$, and make no distinction between TGM and $Hg^0_{air}$, referring to the measurements as $Hg^0_{air}$, when referencing the above-mentioned studies.


The record of hourly mean $Hg^0_{air}$ observations from the four sites used in this study is presented in Fig. 1. Throughout this paper, the data were analysed and are presented in their local time.



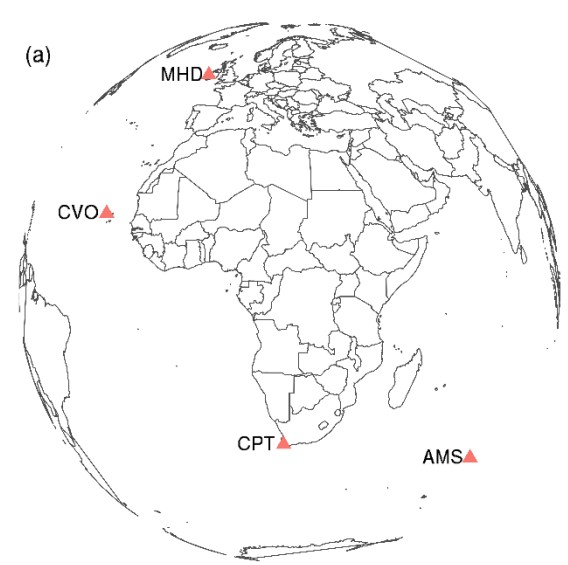

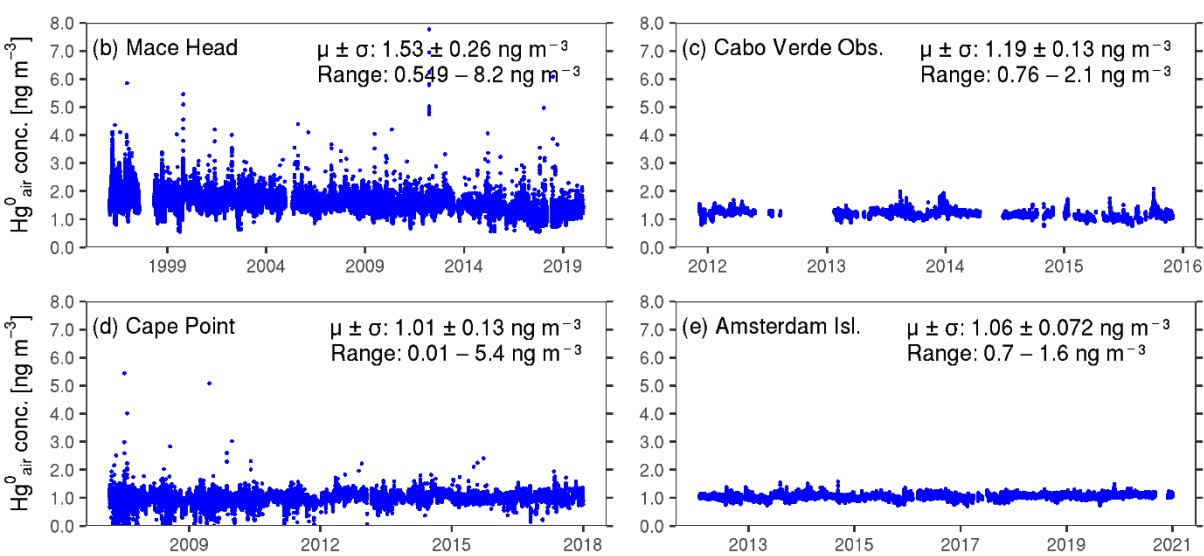

(f)

| Station | Temporal coverage period | No. of data points | Perc. valid data [%] |
| --- | --- | --- | --- |
| Mace Head | 2 Feb 1996–06 Jan 2020 | 171 399 | 82 |
| Cabo Verde Obs. | 5 Dec 2011–29 Dec 2015 | 20 894 | 60 |
| Cape Point | 8 Mar 2007–31 Dec 2017 | 83 303 | 88 |
| Amsterdam Island | 8 Jan 2012–31 Dec 2020 | 56 844 | 73 |

**Figure 1: Overview of the Hg$^0_{air}$ data used in the study. (a) Map showing the locations of the four study sites, Mace Head (MHD), Cabo Verde Observatory (CVO), Cape Point (CPT) and Amsterdam Island (AMS). (b)–(e) Hourly mean Hg$^0_{air}$ concentrations at Mace Head, Cabo Verde Observatory, Cape Point and Amsterdam Island, respectively. The mean ± 1 standard deviation and the range of the hourly observations are shown in each panel. Note that the extent of the x-axis differs across the four sites. For a zoomed-in view of the data, see Fig. S1 in the Supplement. (f) Summary of the hourly mean Hg$^0_{air}$ data coverage at the four sites, showing**





the temporal coverage of the observations used in the present study, the total number of hourly data points and the percentage of hourly time stamps over the coverage period with (valid or non-missing) data.

### 2.1.3 $^{222}$Rn measurements from Cape Point

Since 1999, atmospheric radon ($^{222}$Rn) has been continuously measured at Cape Point using a two flow-loop, dual-filter
detector designed by the Australian Nuclear Science and Technology Organisation (ANSTO) and partially constructed locally (Brunke et al., 2004; Botha et al., 2018). The instrument operates by passing ambient air through a particulate filter into a large chamber, where the $^{222}$Rn gas decays (Brunke et al., 2004). A second filter collects the $^{222}$Rn decay products, which are detected by means of a screen coated with the scintillator material zinc sulphide and viewed by a photomultiplier (Brunke et al., 2004). The count rate is proportional to the ambient $^{222}$Rn concentration (Brunke et al., 2004). The detector at Cape Point has been
continuously developed (Brunke et al., 2004; Botha et al., 2018). In 2011, an upgraded version of the instrument was installed, incorporating system design improvements such as an automated calibration system and a delay volume, increasing the accuracy and reliability of the measurements at the site (Botha et al., 2018).

As described in detail in Botha et al. (2018), the radon detector's sample inlet is located 30 m above ground level and $^{222}$Rn is
measured at a resolution of 30 min and further processed to an average 1 h temporal resolution. The detector is calibrated monthly by introducing a known concentration of $^{222}$Rn from a calibration source ($^{226}$Ra source with $^{222}$Rn production rate of 2.94 Bq min$^{-1}$) acquired from Pylon Electronics, Inc., Canada (Brunke et al., 2004). The instrument has a lower limit of detection of $25 \pm 8$ mBq m$^{-3}$, and the hourly measurements have an uncertainty of around 15 % at 100 mBq m$^{-3}$ and 9 % at 1000 mBq m$^{-3}$ (Botha et al., 2018).


In this work, we use hourly $^{222}$Rn data corresponding to the period of the Cape Point Hg$^0$ dataset (i.e., the period 8 March 2007–31 December 2017).

### 2.2 Air mass back trajectories

We applied the Hybrid Single-Particle Lagrangian Integrated Trajectory (HYSPLIT) model (Stein et al., 2015) to trace air
masses arriving at the monitoring sites. For each station and hour of the day, we calculated a 144-hour (6-day) air mass back trajectory starting at 50 m above model ground level, using the National Centers for Environmental Prediction/National Center for Atmospheric Research (NCEP/NCAR) reanalysis data with a 2.5° spatial and 6-hourly temporal resolution (Kalnay et al., 1996) for model input. Each of the 144 segments of every trajectory was then assigned into one of two categories, based on its geographical location:


- *Marine* (or, *oceanic*): segments located at a bathymetric elevation of 0 m or below.
- *Terrestrial* (or, *continental*): segments located at a bathymetric elevation above 0 m.



The bathymetry information was obtained from the GEneral Bathymetric Chart of the Oceans Grid (GEBCO_GRID), with a
15 arc-second spatial resolution (GEBCO Compilation Group, 2020). The segments were further classified into one of two
categories, according to their height in the atmosphere:

- *Boundary layer*: segments at or below the height of the planetary boundary layer (PBL).
- *Free tropospheric*: segments above the height of the PBL.


For this step, we used PBL height data from the ERA5 re-analysis product of the European Centre for Medium-Range Weather
Forecasts (ECMWF; Hersbach et al., 2020), with a temporal resolution of 1 h and a spatial resolution of 0.25° × 0.25°.
Therefore, each of the 144 segments of every trajectory was assigned to one of four categories: marine boundary layer (MBL),
free troposphere over the ocean (FT_ocean), continental planetary boundary layer (CPBL), or free troposphere over land
(FT_terrestrial; see Fig. S2 in the Supplement).

## 3 Results and discussion

### 3.1 Overview of $Hg^0_{air}$ observations at Mace Head, Cabo Verde Observatory, Cape Point and Amsterdam Island

The record of hourly averaged $Hg^0_{air}$ measurements at the four sites is shown in Fig. 1. Over the analysis periods, the mean
$Hg^0_{air}$ concentration was $1.53 \pm 0.26$, $1.19 \pm 0.13$, $1.01 \pm 0.13$ and $1.06 \pm 0.07$ ng m$^{-3}$ (mean ± 1 standard deviation) at Mace
Head, Cabo Verde Observatory, Cape Point and Amsterdam Island, respectively. The higher mean concentration at Mace Head
and, to a lesser extent, Cabo Verde Observatory compared to Cape Point and Amsterdam Island is consistent with previous
studies that have reported higher $Hg^0_{air}$ levels in the NH than in the SH (e.g., Temme et al., 2003; Soerensen et al., 2010a,
2014; Sprovieri et al., 2010, 2016; Xia et al., 2010). The inter-hemispheric gradient in $Hg^0_{air}$ concentrations has been widely
discussed in the literature and is largely attributed to the bulk burden of anthropogenic Hg emissions occurring in the NH
(Temme et al., 2003; Sprovieri et al., 2010, 2016; Street et al., 2017; AMAP/UNEP, 2018).

Figure 2 shows the seasonal variability of monthly mean $Hg^0_{air}$ concentrations across the four sites. At Mace Head, $Hg^0_{air}$ levels
are slightly but significantly higher in winter (December to May, $1.57 \pm 0.27$ ng m$^{-3}$, n = 84 228) than in summer (June to
November, averaging $1.48 \pm 0.25$ ng m$^{-3}$, n = 87 171) ($p < 2.2e^{-16}$, Mann-Whitney U-Test). As with Mace Head, concentrations
at Cabo Verde Observatory are slightly but significantly higher around winter (December–May, $1.22 \pm 0.12$, n = 11 296)
compared to summer (June–November, $1.16 \pm 0.12$, n = 9 598) ($p < 2.2e^{-16}$, Mann-Whitney U-Test). A similar seasonal pattern
has been observed at other ground-based monitoring stations (e.g., Kock et al., 2005; Lan et al., 2012; Sprovieri et al., 2016;
Jiskra et al., 2018) as well during ship-board sampling campaigns (e.g., Soerensen et al., 2010a) in the NH. Several explanations
have been proposed for the northern hemispheric $Hg^0_{air}$ seasonality, including, increased anthropogenic emissions from fossil-



fuel combustion for heating in winter (Ebinghaus et al., 2002; Lan et al., 2012), higher ocean $Hg^0$ emissions from the North Atlantic Ocean in winter (Soerensen et al., 2010b), and more recently, increased vegetation $Hg^0$ uptake in summer (Jiskra et al., 2018; Feinberg et al., 2022).

Amsterdam Island shows slightly but significantly higher $Hg^0_{air}$ concentrations around winter (June–September, $1.07 \pm 0.06$,
n = 18 697) ($p < 2.2e^{-16}$, Mann-Whitney U-Test). Angot et al. (2014) as well as Slemr et al. (2020), who previously reported the same seasonality for the site, attributed it to long-range transport of polluted air masses from southern Africa between July and September, which coincides with the biomass-burning season in the region. Nevertheless, the observed winter peak at Amsterdam Island (in this study) corresponds to only a ~ 1 % increase from the site's overall mean concentration. Similarly, at Mace Head as well as Cabo Verde Observatory, the winter maximum (summer minimum), though significantly higher
(lower) than the overall mean concentration ($p < 2.2e^{-16}$, Mann-Whitney U-Test), represents only a ~ 3 % change from the overall mean. No clear seasonality is detected at Cape Point.

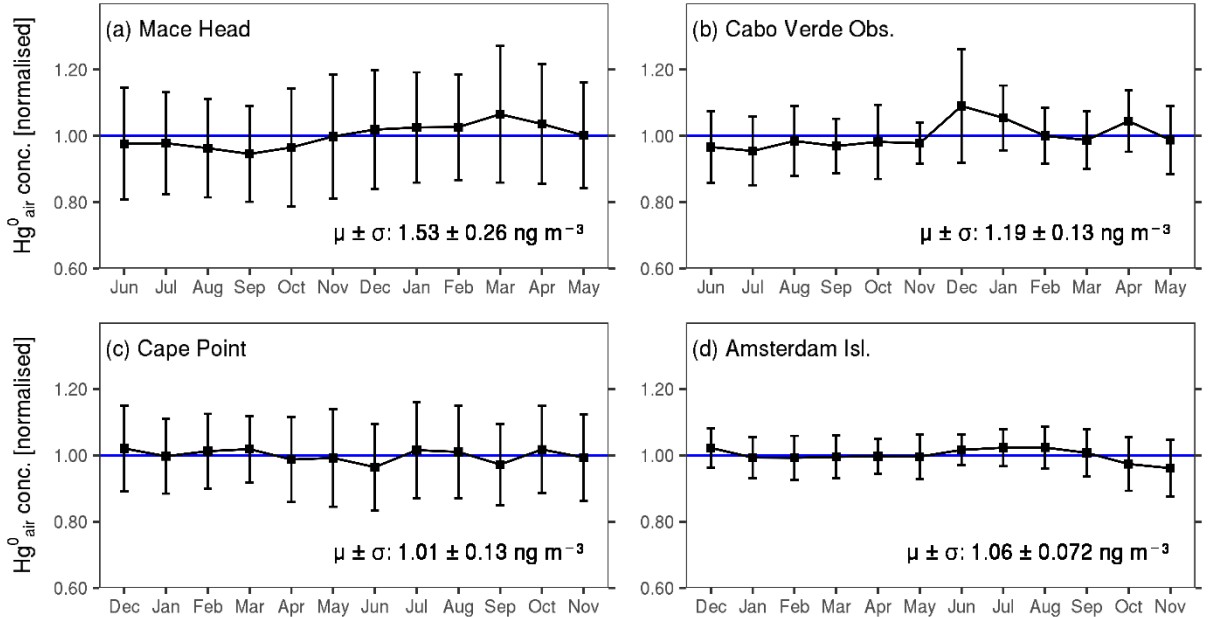

**Figure 2: Normalised monthly mean $Hg^0_{air}$ concentrations at the four sites. The mean ± 1 standard deviation of the hourly**
**observations is shown in each panel. The bars show one standard deviation of the monthly means. The blue horizontal line represents the normalised mean value (i.e., 1 ng m$^{-3}$). The months are ordered starting in June for Mace Head and Cabo Verde Observatory and in December for Cape Point and Amsterdam Island, such that the plots consistently start in summer and end in spring (of the respective hemisphere).**

The diurnal variability of $Hg^0_{air}$ at the four sites is quite weak as well, with daily mean minima and maxima values reflecting just a ~2 % change from the overall mean concentration (Fig. 3). The diurnal patterns also exhibit little seasonal variation (Fig.





S3, Supplement). Non-existent or non-significant $Hg^0_{air}$ diurnal variations in the marine boundary layer (MBL) have also been reported in other studies (e.g, Soerensen et al., 2010a; Wang et al., 2017; Wang et al., 2019).

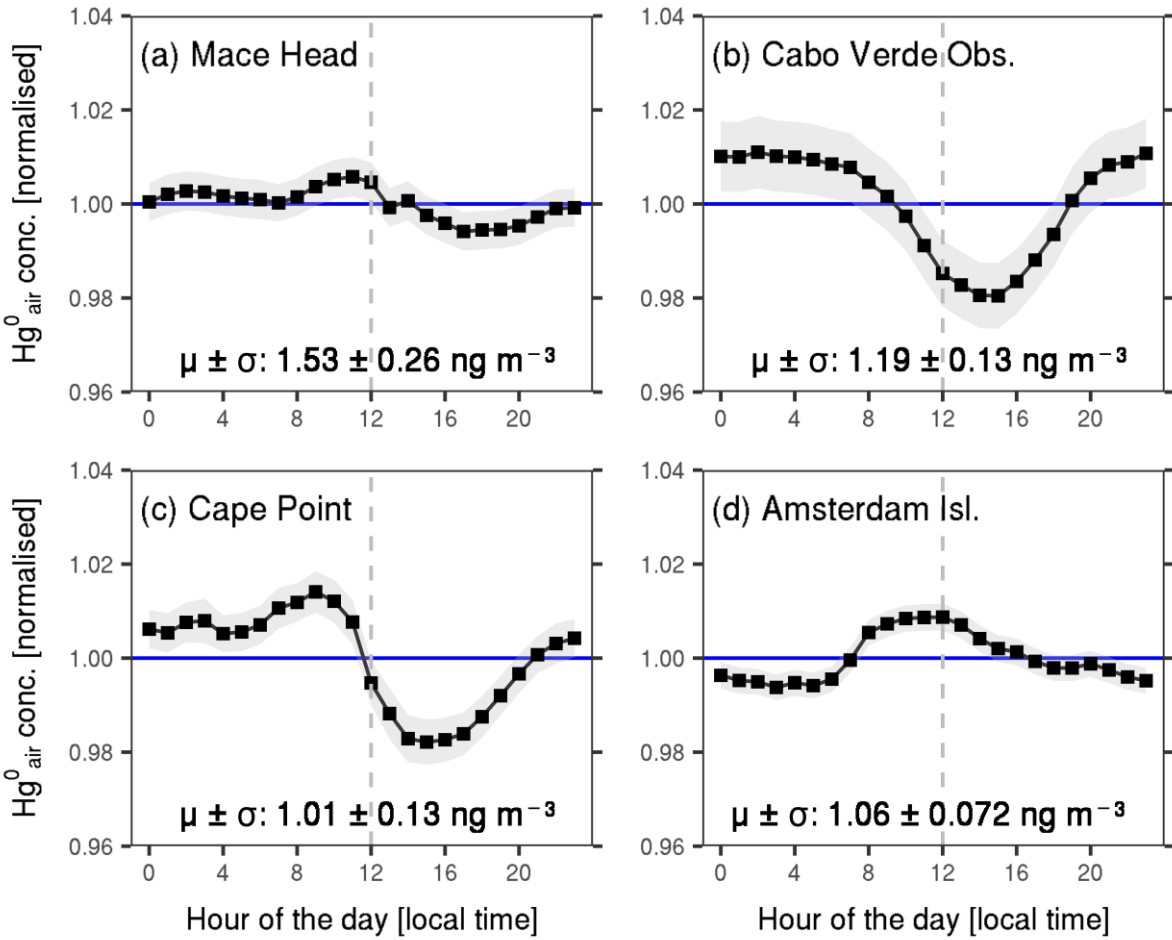

**Figure 3: Normalised mean $Hg^0_{air}$ diurnal variation at Mace Head, Cabo Verde Observatory, Cape Point and Amsterdam Island. The mean ± the standard deviation of the hourly observations is shown for each site. The shaded area shows the mean ± 2 times the standard error. The horizontal line represents the normalised mean value (i.e., 1 ng m⁻³) and the dashed vertical line is plotted along 12:00 local time.**

## 3.2 $Hg^0_{air}$ in the MBL

A major goal of this work was to understand the dynamics of $Hg^0_{air}$ in the MBL, especially given the weak seasonal and diurnal variability in concentrations observed at our ground-based study sites. Previous research has highlighted $Hg^0$ air–sea exchange as one of the key processes influencing $Hg^0_{air}$ concentrations in the MBL (e.g., Wängberg et al., 2001; Laurier et al., 2003;



Soerensen et al., 2010a, 2010b; Carbone et al., 2016; Wang et al., 2017). In turn, $Hg^0$ air–sea exchange is suggested to be driven by two main factors: (i) the $Hg^0$ concentration gradient at the air–sea interface, and (ii) wind speed and solar radiation (e.g., Gårdfeldt et al., 2001; Wängberg et al., 2001; Laurier et al., 2003; Soerensen et al., 2010a, 2010b; 2013; Kuss et al., 2011; Wang et al., 2017). Many studies show that ocean surface waters are primarily supersaturated in $Hg^0_{aq}$ relative to the overlaying air (e.g., Wängberg et al., 2001; Gårdfeldt et al., 2003; Andersson et al., 2011; Mason et al., 2017; Wang et al.,

2017; Wang et al., 2019), creating conducive conditions for net ocean $Hg^0$ emission to the atmosphere. In fact, recent estimates place ocean $Hg^0$ evasion as the largest single net flux of $Hg^0$ to the atmosphere, in the range of 2800–4000 $Hg^0$ t $y^{-1}$ (Horowitz et al., 2017;  AMAP/UNEP, 2018; Outridge et al., 2018; Shah et al., 2021; Zhang et al., 2023). Regarding wind speed and solar radiation, both variables generally exhibit pronounced seasonal and diurnal variability. Considering these factors, it would be expected that $Hg^0_{air}$ concentrations in the MBL also show pronounced seasonal and diurnal variability, but this is not the

case.

While the four sites largely sample clean marine air, they are also occasionally affected by continental/terrestrial sources and processes (e.g., Ebinghaus et al., 2002; Brunke et al., 2004; Slemr et al., 2013, 2020; Angot et al., 2014; Read et al., 2017; Bieser et al., 2020), albeit to varying degrees. As air masses from both the marine and terrestrial environment are sampled at

the sites, this can obscure the link between observed $Hg^0_{air}$ concentrations and the oceanic $Hg^0$ evasion signal. To distinguish between MBL conditions and terrestrial influences, we used the HYSPLIT back trajectories. We classified each hourly $Hg^0_{air}$ observation at the sites into one of three categories: MBL-influenced (MBL_i), terrestrial-influenced (Terr_i) and mixed (Mixed). MBL_i was assigned to observations where 90 % or more of the (144) back trajectory segments were classified as MBL, Terr_i to those where 90 % or more of the segments were classified as CPBL, and mixed was assigned to those where

neither of the first two criteria are met (see Sect. 2.2 for the distinction between MBL- and CPBL-assigned trajectory segments). To validate the classification method, we (also) applied it to $^{222}Rn$ observations from Cape Point. Considering that $^{222}Rn$ emission by the ocean is negligible (at a rate 2 to 3 orders of magnitude lower than the terrestrial flux; Schery and Huang, 2004), and that $^{222}Rn$'s terrestrial source is unaffected by human activity, its consistent emission over land makes it an excellent tracer for identifying air mass of terrestrial origin over the ocean or within the troposphere for  2–3 weeks (Brunke et al., 2010;

Chambers et al., 2018). The $^{222}Rn$ results, presented in Fig. S4 of the Supplement, show generally higher $^{222}Rn$ values for the Terr_i data compared to the MBL_i data, with a significantly larger mean concentration for the Terr_i data (3426 mBq $m^{-3}$, n = 56) compared to the MBL_i data (450 mBq $m^{-3}$, n = 30 470) ($p < 2.2e-16$, Mann-Whitney U-Test). While $^{222}Rn$ measurements are also available at Amsterdam Island, we did not perform the above analysis for the site, as we do not expect a strong terrestrial $^{222}Rn$ signal, due to the remoteness of the site (which is surrounded by over 3000 km of ocean) as well as  the small

size of the island.

Figures S5 and S6 show the seasonal and diurnal variations, respectively, for the MBL_i, Terr_i and Mixed $Hg^0_{air}$ datasets, along with the full observations (All data) at the four sites.  The summary statistics for the groups are provided in Table S1 in



the Supplement. In summary, the results show that the $Hg^0_{air}$ observations at the sites are largely associated with air masses
strongly influenced by the MBL, with little to no contribution from air masses influenced by terrestrial surfaces. The results
also show little difference between the $Hg^0_{air}$ temporal distributions (i.e., the seasonal and diurnal variations) of the full
observations and the MBL_i (as well as the Mixed) data. This suggests that the weak seasonal and diurnal variations observed
at the sites represent the actual MBL conditions and are not an artefact of sampling air masses from different environments.

In a recent publication, Koenig et al. (2023) analysed $Hg^0_{air}$ observations at Maïdo Observatory, a high-altitude station (2160
m above sea level) located on La Réunion Island in the southern tropical Indian Ocean (21°4′ S, 55°48′ E). The study derived
a mean lower free troposphere (LFT) $Hg^0_{air}$ concentration of ~ 0.73 ng m$^{-3}$. Assuming that this value is also representative of
the LFT over Cape Point and Amsterdam Island (located about 3900 and 2900 km from Maïdo observatory, respectively), the
mean MBL (i.e., MBL_i) $Hg^0_{air}$ concentrations extracted at Cape Point and Amsterdam Island ($1.01 \pm 0.12$ and $1.06 \pm 0.07$ ng
m$^{-3}$, respectively; Table S1, Supplement) are both about 30 % larger than the LFT value from Koenig et al. (2023). In the NH,
using aircraft data, Weigelt et al. (2016) reported an LFT $Hg^0_{air}$ concentration of about 1.3 ng m$^{-3}$ over central Europe, which
is about 15 % lower than the mean MBL concentration at Mace Head ($1.53 \pm 0.24$ ng m$^{-3}$). These higher MBL concentrations
relative to LFT concentrations suggest net ocean $Hg^0$ emissions into the MBL.

A question remains, regarding the weak seasonal and diurnal variations in $Hg^0_{air}$ concentrations in the MBL. Another recent
publication, Lamborg et al. (2021), has proposed that dark reduction, rather than photoreduction, as previously believed, is
responsible for most of the $Hg^0_{aq}$ produced in the ocean and subsequently evaded to the atmosphere. Since dark reduction is
less directly impacted by climatic conditions than photoreduction, this would imply constant (i.e., even under low solar
radiation) $Hg^0_{aq}$ supply to the surface ocean. This would have two implications for $Hg^0$ cycling in the air–sea interface. One,
it would explain the weak temporal $Hg^0_{air}$ variability in the MBL, as dark reduction would provide a relatively constant pool
of $Hg^0_{aq}$ to the surface ocean throughout the day and seasons. Secondly, and related to the first point, it would imply that the
ocean is constantly emitting $Hg^0$ to the atmosphere.

With these considerations, we sought to investigate the influence of air mass recent residence time in the MBL on observed
$Hg^0_{air}$ concentrations. We hypothesise that, an air mass entering the MBL from the LFT carries a signature $Hg^0_{air}$ concentration
of this environment. As it travels within the MBL, there is exchange of $Hg^0$ between the air mass and the surface ocean.
Because the ocean surface layer is  typically supersaturated in $Hg^0$ relative to the atmosphere, there is net $Hg^0$ emission to the
atmosphere and consequently, a gradual increase in the $Hg^0$ concentration of the air mass over time. Therefore, air masses
which have recently spent a long time in the MBL would generally have higher $Hg^0_{air}$ levels than those which have recently
spent more time in the LFT. The results of the investigation are presented in the next section.



### 3.3 Relationship between observed $Hg^0_{air}$ concentrations and air mass residence time in the MBL

To test our hypothesis, we used the back trajectories to study the $Hg^0_{air}$ observations in relation to air mass recent MBL residence time. We define air mass recent MBL residence time as the number of hours that an air mass (corresponding to an hourly $Hg^0_{air}$ observation) spent in the MBL in the last 144 hours before reaching the sampling site, that is, the number of

segments of the back trajectory classified as MBL, as per our back trajectory-segment classification (described in Sect. 2.2). We then group together the $Hg^0_{air}$ observations according to air mass recent MBL residence time, at an hourly resolution. Further details on the analysis are provided in Appendix A.

Figure 4 shows mean $Hg^0_{air}$ concentrations against air mass recent MBL residence time across the four sites. As seen in the

figure, $Hg^0_{air}$ concentrations generally increase with air mass recent MBL residence time, in agreement with our hypothesis. The observed increase exhibits an asymptotic pattern, with varying points at which a steady state (i.e., where the concentration remains mostly constant despite increasing time air masses spent in the MBL) is reached across the sites. Air masses sampled at Mace Head, Cabo Verde Observatory, Cape Point and Amsterdam Island reach a mean steady state $Hg^0_{air}$ concentration of about $1.51 \pm 0.02$, $1.16 \pm 0.02$, $1.01 \pm 0.01$ and $1.06 \pm 0.01$ ng m$^{-3}$ (mean $\pm$ standard deviation) at about 60, 72, 30 and 60

hours in the MBL, respectively.

The asymptotic $Hg^0_{air}$ pattern is clearer at Cape Point and Amsterdam Island compared to Mace Head and Cabo Verde Observatory. A possible explanation for this is the contribution from anthropogenic emissions. As already mentioned, overall, anthropogenic Hg emissions are higher in the NH than in the SH. As such, it may be that, at Mace Head and Cabo Verde

Observatory, the anthropogenic (emission) signal is strong enough that it obscures the relationship between $Hg^0_{air}$ concentrations and air mass recent MBL residence time. The fact that at Mace Head and Cabo Verde Observatory the asymptotic $Hg^0_{air}$ pattern is revealed after filtering out air masses which have had recent contact with the continental planetary boundary layer (see Appendix A) supports this hypothesis.

Nevertheless, the observed asymptotic $Hg^0_{air}$ increase at the sites may be partially explained by the exchange of $Hg^0$ between the ocean and the atmosphere. The pattern is consistent with the air–sea gas exchange model (Eq. (1) in Sect. 3.5), which predicts net ocean $Hg^0$ emissions to the surrounding air until the $Hg^0_{air}$ concentration normalised by the Henry's law constant matches the surface ocean $Hg^0_{aq}$ concentration. While other processes may be involved, we are unable to discern them in this study. Given the short travel time considered here (6 days) relative to $Hg^0$'s average atmospheric lifetime against oxidation (6

months–1 year; Horowitz et al., 2017; Saiz-Lopez et al., 2018; Shah et al., 2021), oxidation of $Hg^0_{air}$ is likely insignificant.

We also investigated seasonal variability in the observed relationship between $Hg^0_{air}$ concentrations and air mass recent MBL residence time, however, no clear pattern emerged (Fig. S7, Supplement). Therefore, in the subsequent sections, where we use





the observed relationship to estimate surface ocean $Hg^0_{aq}$ concentrations as well as ocean net $Hg^0$ emissions fluxes, seasonal
analyses are not included.

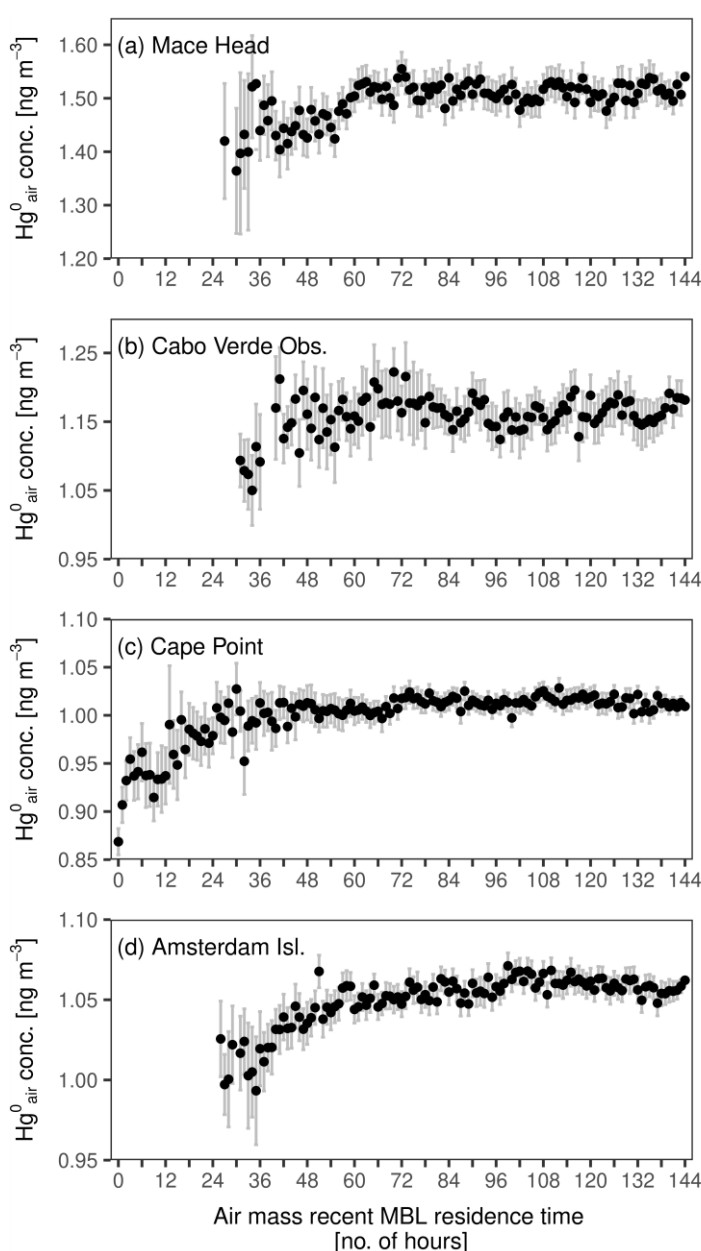

**Figure 4: Mean $Hg^0_{air}$ concentration against air mass recent MBL residence time at the four sites. The bars show 2 times the standard error of the means. Note that the y-axis differs across the four panels.**






**3.4 Estimation of surface ocean $Hg^0_{aq}$ concentrations**

Assuming our findings in Sect. 3.3 can explain most of the variation in $Hg^0_{air}$ concentrations in the MBL, we used the steady state $Hg^0_{air}$ concentrations at the sites to estimate mean surface ocean $Hg^0_{aq}$ concentrations. Based on the back trajectory footprints (see Fig. 5), we assume that the estimates (both the $Hg^0_{aq}$ concentrations, as well as the fluxes, in Sect. 3.5 below)

derived from the Mace Head and Cape Verde Observatory $Hg^0_{air}$ observations represent average conditions for the broad ocean area comprising the North Atlantic and Arctic oceans, while those derived from the Cape Point and Amsterdam Island data represent conditions for the area encompassing the Southern, South Atlantic and south Indian oceans. We apply the equation describing the air–sea exchange flux of $Hg^0$, $F$, based on the thin-film gas exchange model from Liss and Merlivat (1986):


$$F = k_w \left( Hg^0_{aq} - \frac{Hg^0_{air}}{H'} \right), \tag{1}$$

where $k_w$ is the gas transfer velocity, $Hg^0_{aq}$ and $Hg^0_{air}$ are the surface ocean and atmospheric $Hg^0$ concentrations, respectively, and $H'$ is the dimensionless Henry's law constant. At steady-state $F \approx 0$ ng m$^{-2}$ h$^{-1}$ and $Hg^0_{aq}$ is obtained with:


$$Hg^0_{aq} = \frac{Hg^0_{air(ss)}}{H'}, \tag{2}$$

where $Hg^0_{air(ss)}$ is the steady state $Hg^0_{air}$ concentration. $H'$ is calculated using the derivation from Andersson et al. (2008a),

$$H' = e^{\left( \frac{-2404.3}{T} + 6.92 \right)}, \tag{3}$$

where $T$ is the SST. To obtain $T$, we extracted hourly ERA5 SSTs (Hersbach et al., 2020) corresponding to the trajectories associated with the hourly $Hg^0_{air}$ observations in the steady state (i.e., the observations for which air mass recent MBL residence time is greater than or equal to 60, 72, 30 and 60 hours at Mace Head, Cabo Verde Observatory, Cape Point and Amsterdam

Island, respectively). More details are provided in the caption of Table 1, which summarises the $Hg^0_{air(ss)}$, $T$, $H'$ and resultant mean surface ocean $Hg^0_{aq}$ concentration for each site.

From our calculations, we derived a mean surface ocean $Hg^0_{aq}$ concentration of $7.00 \pm 0.21$, $4.00 \pm 0.12$, $4.22 \pm 0.21$ and $4.74 \pm 0.16$ pg L$^{-1}$ (mean $\pm$ 1 standard deviation) based on the $Hg^0_{air}$ steady state concentration at Mace Head, Cabo Verde

Observatory, Cape Point and Amsterdam Island, respectively. The higher estimate for Mace Head compared to Cape Point and Amsterdam Island is in line with elevated anthropogenic Hg emissions and resultant ocean Hg enrichment in the NH relative to the SH. The estimate for Cabo Verde Observatory is lower than that of Mace Head (and closer to that of the two SH sites).



This lower $Hg^0_{aq}$ concentration for Cabo Verde Observatory compared to Mace Head may be attributed to the much higher mean SST corresponding to the average position of the air masses sampled at the former site ($294.89 \pm 0.92$ K at Cabo Verde

Observatory compared to $284.53 \pm 0.92$ K at Mace Head; Table 1). While both sites receive air masses from the North Atlantic, Cabo Verde Observatory also samples tropical air masses, while Mace Head can sample polar air masses (see Fig. 5). Higher SSTs enhance $Hg^0_{aq}$ diffusivity, resulting in lower seawater concentrations.

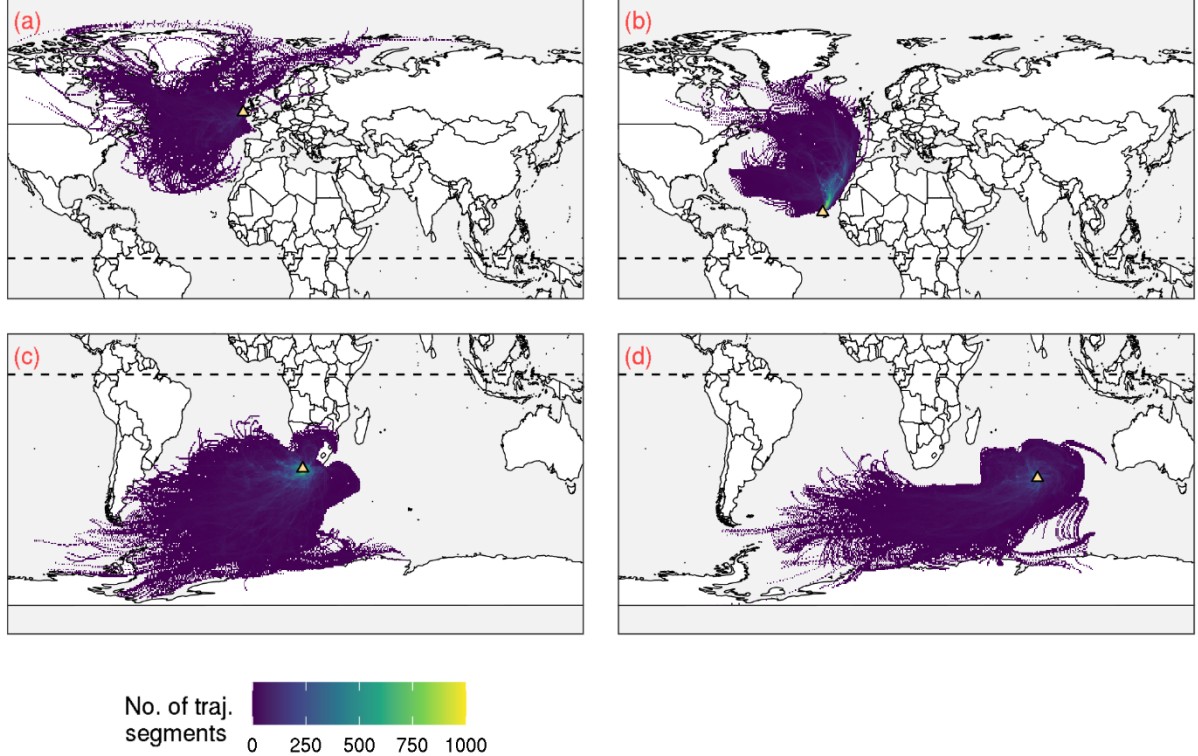

**Figure 5: Back trajectory profiles illustrating the sources of air masses that arrive at (a) Mace Head, (b) Cabo Verde Observatory, (c) Cape Point and (d) Amsterdam Island. The back trajectories correspond to the data considered in deriving the profiles shown in Fig. 4, and shown here is only a subset of the data, from January to December 2013. In each panel, the location of the site is illustrated by the yellow triangle, and the dotted horizontal line shows 0° latitude.**

We compare our estimates to reported surface ocean concentrations in the literature obtained from direct measurements. The estimates from Mace Head and Cabo Verde Observatory are compared to values from the North Atlantic and Arctic oceans, while those from Cape Point and Amsterdam Island are compared to values from the Southern and South Atlantic oceans (no published measurements are available for the Indian Ocean, to the best of our knowledge; see Dastoor et al., 2024). We acknowledge, nevertheless, that our estimates are not directly comparable to these reported measurements for several reasons

(this applies for our evasion flux estimates as well, in the next section). In particular, the concentrations reported in the literature are based on measurements performed over very short periods (days to a few weeks; see Tables 2 and 3, for example), whereas





our estimates are derived from ($Hg^0_{air}$) observations spanning multiple years. Also note that, some of the studies denote the surface water gaseous Hg measurements as dissolved gaseous mercury (DGM; e.g., Gårdfeldt et al., 2003; Wang et al., 2017), the sum of $Hg^0_{aq}$ and dimethylmercury (DMHg). Outside of upwelling regions (Adams et al., 2024), it has been shown that

surface ocean DGM concentrations comprise of mostly $Hg^0_{aq}$, with very minor (if any) contributions from DMHg (e.g., Mason and Fitzerald, 1993; Horvat et al., 2003; Kotnik et al., 2007; Cossa et al., 2011; Bratkič et al., 2016); therefore, we assume that the surface DGM measurements, where they are denoted as such, represent $Hg^0_{aq}$.

**Table 1: Summary (mean ± 1 standard deviation) of steady state $Hg^0_{air}$ (i.e., $Hg^0_{air}$(ss)) concentration, T, H', and resultant surface**
**ocean $Hg^0_{aq}$ concentration across the four sites. For each variable at each site, the mean and standard deviation are based on the mean values of the (steady state) MBL residence time bins. The mean value for each bin is the average of the individual values within the bin. For $Hg^0_{air}$(ss), these are the hourly $Hg^0_{air}$ observations grouped by back trajectory recent MBL residence time. The individual T values are the mean SSTs for the trajectories corresponding to the $Hg^0_{air}$ observations, considering only SSTs associated with segments assigned MBL. These T values are in turn used to obtain H' values, which, together with the $Hg^0_{air}$ concentrations are**
**used to calculate $Hg^0_{aq}$ concentrations according to Eq. (2).**

| Site | $Hg^0_{air(ss)}$ [ng m$^{-3}$] | T [K] | H' | $Hg^0_{aq}$ [pg L$^{-1}$] |
|---|---|---|---|---|
| Mace Head | 1.51 ± 0.02 | 284.53 ± 0.92 | 0.217 ± 0.006 | 7.00 ± 0.21 |
| Cabo Verde Obs. | 1.16 ± 0.02 | 294.89 ± 0.92 | 0.291 ± 0.007 | 4.00 ± 0.12 |
| Cape Point | 1.01 ± 0.01 | 288.03 ± 1.52 | 0.240 ± 0.011 | 4.22 ± 0.21 |
| Amsterdam Isl. | 1.06 ± 0.01 | 285.59 ± 1.08 | 0.224 ± 0.007 | 4.74 ± 0.16 |

Comparison of our $Hg^0_{aq}$ estimates to direct measurements from the literature is presented in Tables 2 and 3 for the NH and SH sites, respectively. As seen in Table 2, there is considerable variability in reported surface ocean $Hg^0_{aq}$ concentrations in the North Atlantic and Arctic oceans. Our Mace Head- and Cabo Verde Observatory-based estimates (7.00 ± 0.21 pg L$^{-1}$ and

4.00 ± 0.12 pg L$^{-1}$, respectively), which lie on the lower end of the reported range, are comparable to the North Atlantic concentrations from Kuss et al. (2011) (in spring, mean of ~ 4–5 pg L$^{-1}$) and Bowman et al. (2014) (10 ± 6 pg L$^{-1}$), as well as the non-ice-covered Arctic Ocean concentrations from DiMento et al. (2019) (6.42 ± 6.01 pg L$^{-1}$), for example. The estimates are rather low compared to the values reported for the Baltic Sea by Osterwalder et al. (2021) (13.5 ± 3.5 pg L$^{-1}$) and Wängberg et al. (2001) (~ 17 pg L$^{-1}$), and especially the Irish west coast (~ 21 pg L$^{-1}$; Gårdfeldt et al., 2003), the western North Atlantic

(mean values ranging between ~ 18 and 39 pg L$^{-1}$; Soerensen et al., 2013) and the Arctic Ocean by Andersson et al. (2008b) (44.13 ± 22.06 pg L$^{-1}$).

The relatively high concentrations reported for the Baltic Sea, Irish west coast and western North Atlantic (from Soerensen et al., 2013), for instance, are (near-)coastal measurements and may not be representative of the open ocean. In contrast, the air

masses sampled at Mace Head and Cabo Verde Observatory spent most of their time over the open ocean, and thus our $Hg^0_{aq}$ estimates reflect open ocean conditions. Coastal waters typically exhibit higher Hg concentrations than the open ocean due to river inputs, which have a much stronger impact in the coastal zone compared to the open ocean (Andersson et al., 2008b;



Soerensen et al., 2013; Amos et al., 2014). Similarly, the Arctic measurements from Andersson et al. (2008b) are also not representative of the non-ice-covered open ocean, due to the capsulating effect of sea ice, which limits air–sea exchange,
leading to the accumulation of $Hg^0_{aq}$ in seawater (Andersson et al., 2008b; DiMento et al., 2019).

For the SH, in situ $Hg^0_{aq}$ measurements are very limited. Nevertheless, for available measurements, our Cape Point (4.22 ± 0.21 pg L$^{-1}$) and Amsterdam Island (4.74 ± 0.16 pg L$^{-1}$) estimates are somewhat comparable to the values reported in the South Atlantic mid-latitudes in autumn (8 ± 3 pg L$^{-1}$, Kuss et al., 2011). The estimates also compare well to the Southern Ocean
measurements reported for summer (7.0 ± 6.8 pg L$^{-1}$) and winter (9 ± 5 pg L$^{-1}$) by Nerentorp Mastromonaco et al. (2017) but are low compared to the summer Southern Ocean measurements from Wang et al. (2017) (24 ± 13 pg L$^{-1}$). Bratkič et al. (2016) report total column DGM measurements for the South Atlantic, with values seeming around 0–10 pg L$^{-1}$ near the surface (See Fig. 6 in Bratkič et al., 2016).

### 3.5 Estimation of ocean net $Hg^0$ evasion fluxes

We also used the observed relationship between $Hg^0_{air}$ concentrations and air mass recent MBL residence time to estimate ocean net $Hg^0$ evasion fluxes. The estimates are derived using the continuity equation in the Lagrangian form describing the change in $Hg^0$ mixing ratio along a trajectory (Brasseur and Jacob, 2017). The detailed methodology is provided in Appendix B.

Based on our calculations, we estimated a mean net evasion flux of 0.75 (95 % confidence interval, CI: 0.34–1.15), 0.58 (0.35–0.82), 0.66 (0.43–0.89) and 0.47 (0.37–0.57) ng m$^{-2}$ h$^{-1}$ for the oceanic airmass source regions for Mace Head, Cabo Verde Observatory, Cape Point and Amsterdam Island, respectively (Fig. B1, Appendix B). Presented in Tables 2 and 3 are these estimates compared to reported values from the literature (calculated using in situ measurements of $Hg^0_{air}$ and surface ocean $Hg^0_{aq}$ concentrations) for the NH and SH sites, respectively.

As with the $Hg^0_{aq}$ concentrations, there is a large variability in reported flux estimates in the North Atlantic and Arctic oceans, with mean evasion fluxes ranging from 0 to around 7 ng m$^{-2}$ h$^{-1}$. Our Mace Head- and Cabo Verde Observatory-based estimates are comparable to values such as those reported for the Baltic Sea in winter (0.83 ng m$^{-2}$ h$^{-1}$; Wängberg et al., 2001) and summer (0.6 ± 06, ng m$^{-2}$ h$^{-1}$; Osterwalder et al., 2021), as well as the North Atlantic mid-latitudes (0.42 ± 0.36 ng m$^{-2}$ h$^{-1}$,
Andersson et al., 2011). The estimates are quite low compared to reported values in Irish west coast (2.7 ng m$^{-2}$ h$^{-1}$; Gårdfeldt et al., 2003) and the western North Atlantic (mean values in the range of ~ 2–7 ng m$^{-2}$ h$^{-1}$; Soerensen et al., 2013).

For the SH, there are only a few flux values in the literature (Table 3). The Cape Point- and Amsterdam Island-based mean estimates from our study are within the range of Southern Ocean values reported by Nerentorp Mastromonaco et al. (2017),



which show marked seasonal variability (-0.2 ± 1.3, 0.4 ± 1.5 and 1.1 ± 1.6 ng m$^{-2}$ h$^{-1}$ in summer, winter and spring, respectively. The estimates are lower than the Southern Ocean value from Wang et al. (2017) (1.5 ± 1.8 ng m$^{-2}$ h$^{-1}$).

**Table 2: Reported surface ocean Hg$^0_{aq}$ concentrations as well as Hg$^0$ air–sea exchange fluxes in the North Atlantic and Arctic oceans obtained from direct measurements, and the estimates from this study based on the relationship between mean Hg$^0_{air}$ concentrations**
**and air mass recent MBL residence time at Mace Head and Cabo Verde Observatory. For the fluxes, positive (negative) values indicate net evasion (deposition).**

| Location | Time period | Hg$^0_{aq}$ concentration [pg L$^{-1}$][a, b] | Hg$^0$ exch. flux [ng m$^{-2}$ h$^{-1}$][a, c] | Reference |
|---|---|---|---|---|
| Baltic Sea | 2–15 Jul 1997 | 17.6 | 1.58 | Wängberg et al. (2001) |
| Baltic Sea | 2–15 Mar 1998 | 17.4 | 0.83 | Wängberg et al. (2001) |
| Irish west coast | Sep 1999 | 21.4 | 2.7 | Gårdfeldt et al. (2003) |
| Artic Ocean | 13 Jul–25 Sep 2005 | 44.13 ± 22.06 | - | Andersson et al. (2008b) |
| N. Atlantic midlatitudes | Nov 2008 | 16 ± 4 | - | Kuss et al. (2011) |
| N. Atlantic subtropics | Nov 2008 | 10 ± 2 | - | Kuss et al. (2011) |
| N. Atlantic midlatitudes | May 2009 | 4 ± 1 | - | Kuss et al. (2011) |
| N. Atlantic subtropics | May 2009 | 5 ± 1 | - | Kuss et al. (2011) |
| North Atlantic | 7–11 Jul 2005 | 11.6 ± 2.0 | 0.42 ± 0.36 | Andersson et al. (2011) |
| N.W. Atlantic | 17–27 Aug 2008 | 32.1 ± 12.0 | 4.3 ± 3.4 | Soerensen et al. (2013) |
| N.W. Atlantic | 21–26 Sep 2008 | 25.7 ± 4.0 | 3.0 ± 2.9 | Soerensen et al. (2013) |
| N.W. Atlantic | 24–27 Jun 2009 | 24.0 ± 4.0 | 4.7 ± 3.7 | Soerensen et al. (2013) |
| N.W. Atlantic | 31 Aug–4 Sep2009 | 22.3 ± 2.8 | 2.1 ± 0.7 | Soerensen et al. (2013) |
| N.W. Atlantic | 29 Sep–7 Oct 2009 | 18.1 ± 4.8 | 2.2 ± 1.7 | Soerensen et al. (2013) |
| N.W. Atlantic | 4–9 Aug 2009 | 39.3 ± 6.8 | 6.8 ± 5.1 | Soerensen et al. (2013) |
| N. Atlantic | Autumn 2010, 2011 | 10 ± 6[d] | - | Bowman et al. (2014) |
| Eastern N. Atlantic | Autumn 2010, 2011 | 9.4 ± 6.01[d] | 0.60 ± 0.67 | Mason et al. (2017) |
| Western N. Atlantic | Autumn 2010, 2011 | 11 ± 7.4[d] | 1.23 ± 1.56 | Mason et al. (2017) |
| Arctic Ocean | 9–12 Oct 2015 | 6.42 ± 6.01[e] | 0.4 ± 2.8[e] | DiMento et al. (2019) |
| Arctic Ocean | 9–12 Oct 2015 | 20.26 ± 19.65[f] | 2.8 ± 10.43[f] | DiMento et al. (2019) |
| Baltic Sea | 10 May–20 Jun 2017 | 13.5 ± 3.5 | 0.6 ± 0.6 | Osterwalder et al. (2021) |
| Mace Head | Feb 1996–Jan 2020 | 7.00 ± 0.21 | 0.75 (0.34–1.15)[g] | Present study |
| Cabo Verde Obs. | Dec 2011–Dec 2015 | 4.00 ± 0.12 | 0.58 (0.35–0.82)[g] | Present study |

[a] Mean ± standard deviation (if standard deviation given), except for our flux estimates, where we report the mean and the upper and lower bounds of the 95 % confidence interval

[b] Some concentrations reported in ng m$^{-3}$, fM or pM, and converted here to pg L$^{-1}$ as follows: 1 ng m$^{-3}$ = 1 pg L$^{-1}$, 1 fM = 0.20059 pg L$^{-1}$ and 1 pM = 200.59 pg L$^{-1}$.

[c] Some fluxes reported in ng m$^{-2}$ d$^{-1}$, pmol m$^{-2}$ h$^{-1}$ or pmol m$^{-2}$ d$^{-1}$, and converted here to ng m$^{-2}$ h$^{-1}$ as follows: 1 ng m$^{-2}$ d$^{-1}$ = 1/24 ng m$^{-2}$ h$^{-1}$, 1 pmol m$^{-2}$ h$^{-1}$ = 0.20059 ng m$^{-2}$ h$^{-1}$ and 1 pmol m$^{-2}$ d$^{-1}$ = 0.20059/24 ng m$^{-2}$ h$^{-1}$.

[d] Values from the same sampling campaign.

[e] Value in ice-free open waters; [f] value in contiguous ice-covered waters.

[g] Mean and the upper and lower bounds of the 95 % confidence interval of the mean.





**Table 3: Similar Table 2, but for reported values in the South Atlantic and Southern oceans, and estimates from this study for Cape Point and Amsterdam Island. Here, all reported Hg⁰ₐq concentrations are surface ocean values except those marked with \*, which**
**are total column concentrations. For the fluxes, positive (negative) values indicate net evasion (deposition).**

| Location | Time period | $Hg^0_{aq}$ concentration [pg L$^{-1}$][a, b] | $Hg^0$ exch. flux [ng m$^{-2}$ h$^{-1}$][a] | Reference |
|---|---|---|---|---|
| S. Atlantic subtropics | Apr 2009 | 9 ± 2 | - | Kuss et al. (2011) |
| S. Atlantic midlatitudes | Apr 2009 | 8 ± 3 | - | Kuss et al. (2011) |
| S. Atlantic | 24 Dec 2011–27 Jan 2012 | 45 ± 29* | - | Bratkič et al. (2016) |
| Southern Ocean | 8 Jun–12 Aug 2013 | 9 ± 5 | 0.4 ± 1.5 | N-Mastromonaco et al. (2017) |
| Southern Ocean | 14 Aug–16 Oct 2013 | 12 ± 7 | 1.1 ± 1.6 | N-Mastromonaco et al. (2017) |
| Southern Ocean | 8 Dec 2010–14 Jan 2011 | 7 ± 6.8 | -0.2 ± 1.3 | N-Mastromonaco et al. (2017) |
| Southern Ocean | 13 Dec 2014–1 Feb 2015 | 24 ± 13 | 1.5 ± 1.8 | Wang et al. (2017) |
| Cape Point | Mar 2007–Dec 2017 | 4.22 ± 0.21 | 0.66 (0.43–0.89)[c] | Present study |
| Amsterdam Island | Jan 2012–Dec 2020 | 4.74 ± 0.16 | 0.47 (0.37–0.57)[c] | Present study |

[a] Mean ± standard deviation (if standard deviation given), except for our flux estimates, where we report the mean and the upper and lower bounds of the 95 % confidence interval.

[b] Some concentrations reported in ng m$^{-3}$ and converted here to pg L$^{-1}$ as follows: 1 ng m$^{-3}$ = 1 pg L$^{-1}$.

[c] Mean and the upper and lower bounds of the 95 % confidence interval of the mean.

## 3.6 Global extrapolation

By extrapolation of our site-specific flux results, we estimated a global mean evasion flux. For this, we assume that the mean of our Mace Head- and Cabo Verde Observatory-based estimates, 0.67 ng m$^{-2}$ h$^{-1}$ (95 % CI: 0.35–0.99 ng m$^{-2}$ h$^{-1}$), is
representative of the North Atlantic and Arctic oceans (combined area: ~ 57.048 million km$^2$), and the mean of the Cape Point- and Amsterdam Island-based estimates, 0.57 ng m$^{-2}$ h$^{-1}$ (0.40–0.73 ng m$^{-2}$ h$^{-1}$), is representative of the Southern, South Atlantic and Indian oceans (~ 132.79 million km$^2$). For the Pacific Ocean (~ 168.723 million km$^2$), which is not covered by our observatories, we use the mean of the four estimates, 0.62 ng m$^{-2}$ h$^{-1}$ (0.37–0.86 ng m$^{-2}$ h$^{-1}$). Under these assumptions, we obtain a global mean evasion flux of around 1900 t y$^{-1}$, with a 95 % confidence interval of 1200–2600 t y$^{-1}$. This flux is rather
small compared to recent model estimates, which range from 2800 to 4000 Hg$^0$ t y$^{-1}$ (Horowitz et al., 2017;  AMAP/UNEP, 2018; Outridge et al., 2018; Shah et al., 2021; Zhang et al., 2023), suggesting that net oceanic Hg$^0$ emission may currently be overestimated. Such an overestimation may relate to the parameterisation of the air–sea exchange velocity (e.g., Nightingale et al., 2000; McGillis et al., 2001) used in the models which is a large source of uncertainty (Zhang et al., 2019; Osterwalder et al., 2021).


That being said, our estimate is evidently subject to significant uncertainties, particularly due to the strong assumption that the results corresponding to individual sites and their oceanic areas of influence (see Fig. 5) are somewhat representative of larger oceanic regions. In addition, back trajectories, on which our estimate is based, provide only a general indication of air mass



source regions, and are prone to large uncertainties from possible errors in input meteorological fields as well as numerical
models (Yu et al., 2009).

**4 Summary and conclusions**

In this study, we combined long-term $Hg^0_{air}$ observations with HYSPLIT air mass back trajectories at four ground-based
monitoring sites, Mace Head and Cabo Verde Observatory in the NH, and Cape Point and Amsterdam Island in the SH, to
study $Hg^0$ air–sea exchange. At all sites, we observed weak seasonal and diurnal $Hg^0_{air}$ variations, even after filtering the data
to extract observations representative of MBL conditions. This suggests that the weak variations reflect actual MBL dynamics
and are not an artefact of the sampling of air masses of both oceanic and continental origin.

We investigated the relationship between $Hg^0_{air}$ concentrations and the recent residence time of air masses in the MBL. Our
results showed a gradual increase in mean $Hg^0_{air}$ concentration with air mass recent MBL residence time, followed by a steady
state. The observed pattern is consistent with the thin film gas exchange model, which predicts net ocean $Hg^0$ emissions into
the atmosphere until the $Hg^0_{air}$ concentration normalised by the Henry's law constant matches the surface ocean $Hg^0_{aq}$
concentration. This provides strong evidence that ocean $Hg^0$ emissions directly influence the observed $Hg^0_{air}$ concentrations at
the sites. Combined with the weak seasonal and diurnal variability at the sites, our findings suggest that recent air mass MBL
residence time has a stronger influence on $Hg^0_{air}$ concentrations in the MBL than seasonal and diurnal factors, such as solar
radiation.

Using the relationship between mean $Hg^0_{air}$ concentrations and air mass recent MBL residence time at the sites, we estimated
surface ocean $Hg^0_{aq}$ concentrations as well as ocean net $Hg^0$ evasion fluxes for the North Atlantic and Arctic oceans (AA) and
the Southern, South Atlantic and south Indian oceans (SSI). The estimated $Hg^0_{aq}$ concentration was in the range of ~ 4–7 pg
$L^{-1}$ for the AA and around 4 pg $L^{-1}$ for the SSI, while the mean evasion flux was around 0.6–0.8 ng $m^{-2}$ $h^{-1}$ and 0.5–0.7 ng $m^{-2}$
$h^{-1}$ for the AA and SSI, respectively. Extrapolating the fluxes to the global scale, we derived a global net $Hg^0$ evasion flux of
around 1900 t $y^{-1}$ (95 % CI: 1200–2600 t $y^{-1}$). While this global estimate is based on strong assumptions and is subject to
significant uncertainties, it suggests that current model estimates of net $Hg^0$ evasion flux (2800–4000 $Hg^0$ t $y^{-1}$) may be too
high. To improve the accuracy of the flux estimate, it is crucial to conduct longer term, higher frequency in situ measurements
of $Hg^0$ air–sea exchange fluxes (as well as $Hg^0_{aq}$ concentrations). These efforts are particularly crucial in the SH, where direct
measurements are currently very sparse. Such comprehensive measurements would significantly improve our understanding
and modeling of Hg cycling in the MBL.

The methodology applied here could be improved. In particular, the air mass back trajectories could be computed with higher
spatially and temporally resolved meteorological input, if enough computational resources are available. However,  while



higher-resolution input data could provide more accurate insights into the sources and travel paths of air masses, a relatively low-resolution input data (NCEP/NCAR Reanalysis) was chosen here considering the large number of trajectories (e.g., every hour between 1996 and 2020 for Mace Head) which had to be computed for the study.

While the results of this paper are subject to significant uncertainties, our study demonstrates the applicability of ground-based $Hg^0_{air}$ observations, where the currently available measurements span multiple years, in studying air–sea exchange as well constraining surface ocean $Hg^0_{aq}$ concentrations and net ocean $Hg^0$ evasion fluxes.

**Appendix A. $Hg^0_{air}$ concentration in relation to air mass recent MBL residence time**

We used the HYSPLIT air mass back trajectories to investigate the relationship between observed $Hg^0_{air}$ concentration and air
mass recent MBL residence time. Air mass recent MBL residence time is defined as the number of hours that an air mass (corresponding to an hourly $Hg^0_{air}$ observation) spent in the MBL in the past 144 hours before reaching the site, i.e., the number of segments of the back trajectory that are classified as MBL, as per our back trajectory segment categorisation (described in Sect. 2.2). We then group together the $Hg^0_{air}$ observations according to recent air mass MBL residence time, at an hourly temporal resolution.


Figure A1 shows the mean $Hg^0_{air}$ concentrations against air mass recent MBL residence time at the four sites. The left-hand panels show the initial results. For Cape Point and Amsterdam Island (Figs. A1(g) and A1(i), respectively), there is a clear pattern of increasing mean $Hg^0_{air}$ concentration with recent air mass MBL residence time, albeit with some outliers. At Mace Head, we observe an increase in mean $Hg^0_{air}$ concentration from 0 to about 14 hours in the MBL, followed by a decline until
about 48 hours, a slight increase, and then steady concentrations (Fig. A1(a)). Cabo Verde Observatory shows a similar pattern, with decreasing concentrations until about 60 hours of MBL duration, followed by an increase until about 68 hours and then mostly steady $Hg^0_{air}$ concentrations (Fig. A1 (d)).

We expected the pattern observed at Cape Point and Amsterdam Island to be present at Mace Head and Cabo Verde
Observatory as well, so we investigated further. Since anthropogenic Hg emissions are generally higher in the NH compared to the SH, we hypothesised that, at Mace Head and Cabo Verde Observatory, the relationship between mean $Hg^0_{air}$ concentrations and air mass recent residence time in the MBL is obscured by anthropogenic emissions. This would explain the relatively high mean $Hg^0_{air}$ concentrations at low MBL residence times at the two NH sites (Figs. A1(a) and A1(d)). To reduce the impact of anthropogenic emissions, we removed the hourly $Hg^0_{air}$ observations where the air mass recently had contact
with the terrestrial surface, i.e., where the corresponding air mass back trajectory has any segments classified as CPBL (see Fig. A2 for an illustration of the of the trajectory filtering). These results, referred to as No_CPBL, are shown in the middle column of Fig. A1. We tested various (less strict) versions of this filter, for instance, removing the $Hg^0_{air}$ observations where





the back trajectory had a certain percentage of segments in the CPBL only in the last 6, 12, or 24 hours before arrival at the site (not shown). These various filters produced more or less similar results to the No_CPBL filter.


As seen in Fig. A1(b) and A1(e), reducing the influence of the terrestrial environment at Mace Head and Cabo Verde Observatory, respectively, reveals a similar pattern to that observed at the SH sites, albeit with considerable noise at the lower MBL residence time groups, particularly at Mace Head. Further investigation showed that that the noisy values were due to a low number of observations in the groups (in some instances, a single data point). Therefore, we applied a 12-data-points

(equivalent to half a day's observations) minimum filter, removing groups with fewer than 12 observations. This rule was applied to all sites, and the results (Minimum data points) are presented on the right-hand panels of Fig. A1.



**Figure A1: Mean Hg$^0_{air}$ concentration against air mass recent MBL residence time at Mace Head, Cabo Verde Observatory, Cape Point and Amsterdam Island, showing the progressive filtering applied. In the panel on the left ("All data"), the entire data set is used at each site. In the middle panels ("No_CPBL"), Hg$^0_{air}$ points in which the corresponding back trajectory has any segments categorised as CPBL have been removed; this is applied only at Mace Head and Cabo Verde Observatory. In the panels on the right ("Minimum data points"), the same data to the left is used but MBL residence times with less than 12 Hg$^0_{air}$ observations are removed. The bars show 2 times the standard error of the means. Note that the extent of the y-axis differs across the four stations.**

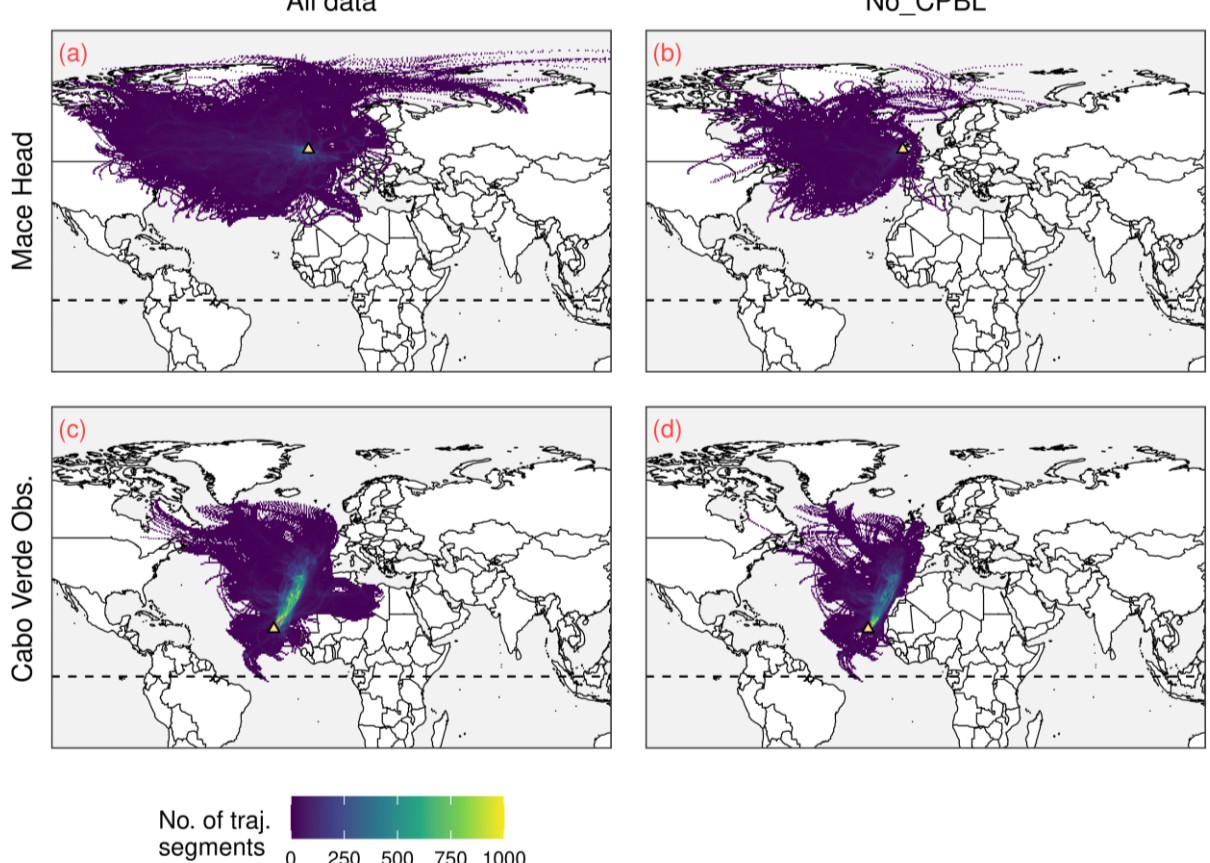

**Figure A2: Illustration of the back trajectory-based filtering applied at Mace Head and Cabo Verde Observatory, with a sample from Jul–Dec 2013. The left-side panels ((a) and (c)) show the trajectories corresponding to the whole data, and on the right ((b) and (d)) the trajectories which have any segments categorised as CPBL have been removed. The dashed horizontal line shows 0° latitude and the location of the station is illustrated with the yellow triangle.**

*Influence of Agulhas Current region emissions at Amsterdam Island*

The Hg$^0_{air}$ observations at Amsterdam Island show a clear pattern of increasing concentration with air mass recent MBL residence time (Fig. A1(i)–(j)). Nevertheless, there are occurrences of relatively high mean concentrations at low MBL residence time groups. We investigated these occurrences, by studying the individual hourly observations in these groups. The



investigation showed that for most of these occurrences, one or more of the hourly observations has an unusually high value (considering the site's mean of $1.06 \pm 0.07$ ng m$^{-3}$), over 1.3 ng m$^{-3}$. Further investigation into these unusually high concentrations showed a close association with air masses coming from the Agulhas Current region. This finding is analogous

to that of Bieser et al. (2020), who observed that elevated Hg$^0_{air}$ concentrations at Cape Point were linked to air masses from the warm Agulhas Current region. We therefore remove all hourly observations at Amsterdam Island where the corresponding trajectory has travelled over the Agulhas Current region (Fig. A3). For Cape Point, we did not observe the same impact, hence the same filtering was not applied.


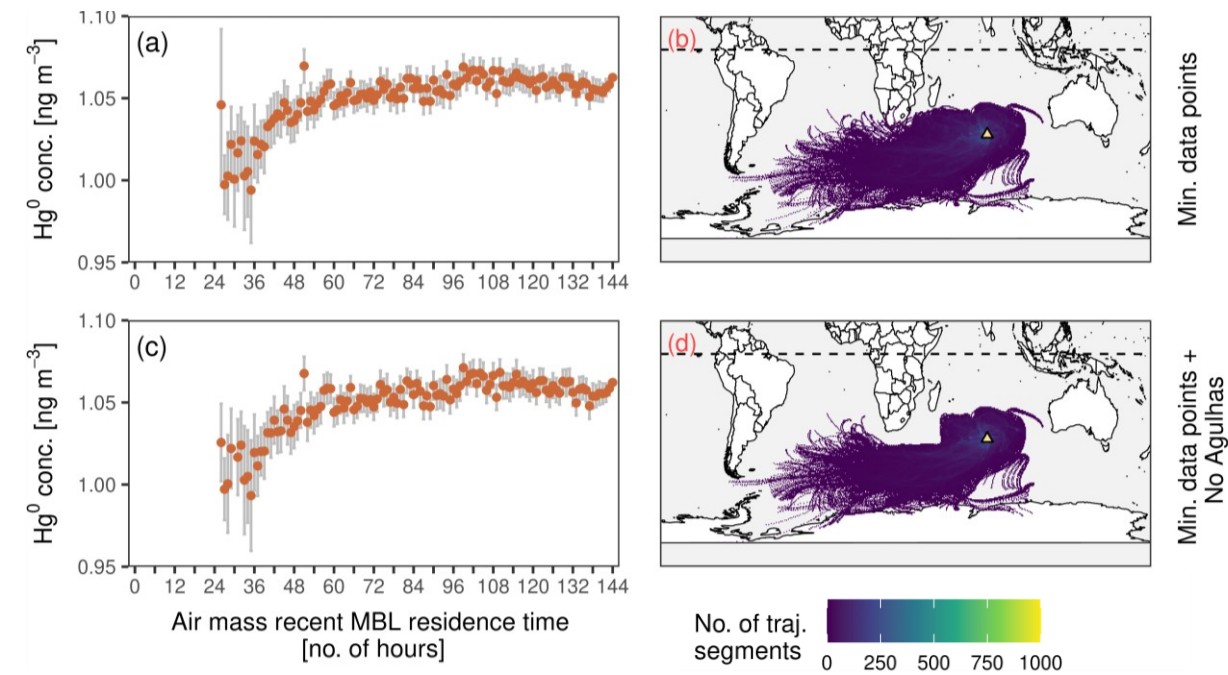

**Figure A3: Illustration of the Agulhas Current filtering applied at Amsterdam Island. (a)–(c) Mean Hg$^0_{air}$ concentration against air mass recent MBL residence time, and ((b)–(d)) the corresponding air mass back trajectories (a sample, Jan–Jun 2013). (a) Is the same as Fig. S7(j), while in (c) Hg$^0_{air}$ observations where the back trajectory has travelled over the Agulhas Current (defined as**
**latitude between 40° S and 27° S and longitude between 20° E and 50° E) have been removed (as shown in (d)). In panels (a) and (c) the bars show 2 times the standard error of the means. On the maps the location of the station is shown by the yellow triangle and the dotted horizontal line shows 0° latitude.**

## Appendix B. Constraining ocean net Hg$^0$ evasion fluxes from the observed relationship between mean Hg$^0_{air}$ concentration and air mass recent MBL residence time

We use the observed relationship between mean Hg$^0_{air}$ concentration and air mass recent MBL residence time at the sites to constrain the associated mean ocean net Hg$^0$ evasion fluxes. We apply the continuity equation in the Lagrangian form describing the change in Hg$^0$ mixing ratio along a trajectory (Brasseur and Jacob, 2017):



$$\frac{d\mu}{dt} = \frac{1}{\rho_a}\frac{F}{h_{PBL}}, \tag{B1}$$


where $\mu = c/\rho_a$ is the mass mixing ratio of $Hg^0$ in air, $c$ is the $Hg^0$ air concentration and $\rho_a$ the air density, $h_{PBL}$ is the planetary boundary layer height and $F$ is the air-sea $Hg^0$ exchange flux. The flux $F$ is expressed according to:

$$F = k_w\left(c_w - \frac{c}{H'}\right), \tag{B2}$$


where $k_w$ is the gas transfer velocity, $c_w$ is the seawater dissolved $Hg^0$ concentration and $H' = c/c_w$ is the dimensionless Henry's law constant.

Substituting Eq. (B2) into Eq. (B1) and multiplying the resulting equation by air density $\rho_{st} = 1.275$ kg m$^{-3}$ at standard

temperature and pressure ($T_{st} = 273.15$ K, $p_{st} = 10^5$ Pa, IUPAC), we obtain the equation for $Hg^0$ air concentration at standard temperature and pressure (as measured by Tekran):

$$\frac{dc}{dt} = B - Ac, \tag{B3}$$

where $A = \frac{k_w}{H'h_{PBL}}$ and $B = \frac{\rho_{st}}{\rho_a}\frac{k_w c_w}{h_{PBL}}$.

Assuming $A$ and $B$ are constants in the first approximation, Eq. (B3) has a solution:

$$c(t) = \frac{B}{A} - \left(\frac{B}{A} - c_0\right)e^{-At}, \tag{B4}$$


where $c_0$ is the initial concentration at $t = 0$.

Equation (B4) can be transformed to

$c(t) = c_{eq} - De^{-At}, \tag{B5}$

where $c_{eq} = B/A$ is the equilibrium concentration ($t \rightarrow \infty$), and $D = c_{eq} - c_0$.



Equation (B5) describes the temporal change in the $Hg^0$ concentration (at standard temperature and pressure) in an air parcel
moving along a trajectory.

We obtain $c_{eq}$ by averaging the $Hg^0{}_{air}$ concentrations at the steady state. Shown in the left-most panels of Fig. B1 are the $c_{eq}$
values (which are also described in Sect. 3.3 of the main text). Next, taking the natural logarithm of $c_{eq}$–$c$, we obtain from Eq.
(B5):


$$ln\left(c_{eq} - c\right) \;=\; lnD - At. \tag{B6}$$

Plotting $ln(c_{eq}$–$c)$ against air mass MBL residence time and applying a linear approximation, we obtain $A$ and $lnD$, as well as
the standard error (SE) of $A$ (see middle panels of Fig. B1). Using the $A$ and $lnD$, we obtain the final approximation of $c(t)$, as
shown in the left-most panels of Fig. B1.







**Figure B1: Sequential process of estimating mean ocean net Hg⁰ evasion fluxes from the relationship between mean Hg⁰ₐᵢᵣ concentration and air mass recent MBL residence time at the monitoring sites.** Left-most panels: Mean Hg⁰ₐᵢᵣ concentration against
air mass recent MBL residence time (black dots), with the mean steady state Hg⁰ₐᵢᵣ concentration ($c_{eq}$) and the approximation for $c(t)$ (curve and equation) also shown. The shading gives the approximate 95 % confidence interval of $c(t)$ (derived using $A \pm 2$ times SE of $A$). Middle panels: $ln(c_{eq}−c)$ against air mass recent MBL residence time. Also illustrated is the linear approximation of $ln(c_{eq}−c)$ (curve and the equation) as well as the resultant $A$ (including the SE of $A$, in brackets) and $lnD$ estimates. Right-most panels: Ocean net Hg⁰ evasion flux ($F$) against air mass recent MBL residence time. The three curves correspond to the best approximation
and the upper and lower bounds of the 95 % confidence interval of $F$ (calculated using $A \pm 2$ times SE of $A$). The mean values of the curves, $\mu_F$, are shown on the top right.

*Flux estimations using the relationship between mean Hg⁰ₐᵢᵣ concentrations and air mass recent MBL duration*

Substituting expressions for $A$ and $B$ from Eqs. (B3) into Eq. (B2) and taking into account that $C_{eq} = B / A$, we derive:



$$F = h_{PBL} A \left( \frac{\rho_a}{\rho_{st}} c_{eq} - c \right). \tag{B7}$$

Assuming $\rho_a \approx \rho_{st}$ we finally obtain:


$$F \approx h_{PBL} A \left( c_{eq} - c \right), \tag{B8}$$

where $h_{PBL}$ is the mean (ERA5-derived) planetary boundary layer height corresponding to the trajectories associated with the Hg$^0_{air}$ observations, considering only the trajectory segments assigned as MBL. The final approximation of $F$ is shown in right-

most panels of Fig. B1. We assume that the coloured curves in Fig. B1 (left column) show the change of Hg$^0$ concentrations during the recent (last 144 h) transport history of a typical air parcel travelling over the oceanic regions represented by our sites and obtain the mean evasion flux by averaging the instantaneous evasion flux shown in Fig. B1 (right column).

*Data availability*. The Amsterdam Island L2 Hg$^0_{air}$ data (https://doi.org/10.25326/168) are freely accessible from the GMOS-

FR website https://gmos.aeris-data.fr/ (Magand and Dommergue, 2021). The Cabo Verde Observatory Hg$^0_{air}$ data are publicly available following free registration at the National Centre for Atmospheric Science (CEDA) Archive https://catalogue.ceda.ac.uk/uuid/0ae5eb7ce3ad4885a7223dd7b69f4db6/ (Read et al., 2017). Hg$^0_{air}$ data from Mace Head and Cape Point can be obtained upon request from https://gos4m.org. The Cape Point $^{222}$Rn data is publicly accessible following free registration from the World Data Centre for Greenhouse Gases website https://gaw.kishou.go.jp/. The HYSPLIT model

and the NCEP/NCAR reanalysis used as input are publicly available from https://www.ready.noaa.gov/HYSPLIT.php (Stein et al., 2015) and ftp://arlftp.arlhq.noaa.gov/pub/archives/reanalysis/ (Kalnay et al., 1996), respectively. ERA5 data are publicly accessible from the Copernicus Climate Data Store (https://cds.climate. copernicus.eu/) (Hersbach et al., 2020). The GEBCO gridded bathymetry are freely available at https://www.gebco.net/data_and_products/historical_data_sets/#gebco_2020 (GEBCO Compilation, 2020).


*Author contributions*. KMM performed formal analysis, created the visualisations, and wrote the manuscript with contributions from all co-authors. KMM, JB, AMK, IMH, AD, OM, HA, OT and RE contributed to experimental and statistical methods. OM, RE, HA, YB and LM were involved in data curation. OM performed field work and validated the experimental setup. JB, IMH, RE, AD, OT, LM and KR acquired funding. JB, IMH, AD and RE supervised the research.


*Competing interests*. One co-author is a member of the editorial board of Atmospheric Chemistry and Physics.

*Acknowledgements*. We thank the Irish Environmental Protection Agency, through its Climate Change Research Programme, for the support of the Hg measurement programme at Mace Head.




The Cabo Verde Observatory Hg measurements were supported by infrastructure and ancillary measurements funded through the Atmospheric Measurement and Observation Facility (AMOF, part of the National Centre for Atmospheric Science, NCAS) in the UK.

The atmospheric Hg measurements made at Cape Point have been supported by the South African Weather Service (SAWS) and have also received financial support from the Global Mercury Observation System (GMOS), a European Community-funded FP7 project (ENV.2010.4.1.3-2) and from the Department of Science and Innovation as part of the SA Mercury Network (SAMNet). The Cape Point Team is also grateful for the support of the Radon measurement program at the station, in particular Dr Alastair Williams from ANSTO in Australia.


We thank all overwintering staff at Amsterdam Island, as well as the staff and scientists from French Polar Institute Paul-Emile Victor (IPEV), for their assistance with the setup and maintenance of the experiment at Amsterdam Island in the framework of the GMOStral-1028 IPEV program from 2012. Amsterdam Island atmospheric Hg data, accessible from the national GMOS-FR website data portal (https://gmos.aeris-data.fr/), were collected via instruments coordinated by the IGE-PTICHA
technical platform dedicated to atmospheric chemistry field instrumentation. The GMOS-FR data portal is maintained by the French national centre for Atmospheric data and services, AERIS (https://www.aeris-data.fr/), which is acknowledged by the authors. The authors would also like to acknowledge the contribution received from the following projects: Amsterdam Island Hg data, accessible in GMOS-FR national database (https://gmos.aeris-data.fr/), has been collected with funding from the European Union 7th Framework Programme project, GMOS (GMOS 2010-2015), the French Polar Institute (IPEV) via
GMOStral-1028 IPEV program since 2012, the LEFE CHAT CNRS/INSU (TOPMMODEL program) and the H2020 ERA-PLANET (689443) iGOSP programme.

Lastly, we extend our gratitude to Danilo Custódio, Jenny Fisher and Jeroen Sonke for their discussions at various stages of this research output.


*Financial support.* This work is part of the GMOS-Train project that has received funding from the European Union's Horizon 2020 research and innovation programme under Marie Skłodowska-Curie grant agreement no. 860497.

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
