# Peer review of "Constraining elemental mercury air—sea exchange using long-term ground-based observations"

_EGUsphere, 2024_

## Author Comment (AC2)

Response to Anonymous Reviewer #1

Molepo et al. examined long-term measurements of atmospheric Hg(0) at four coastal or oceanic island sites, focusing on steady-state air Hg(0) concentrations in the MBL over global oceans. Based on these findings and a previous air-sea thin film exchange model, the authors sought to calculate mean dissolved Hg(0) concentrations in surface seawater and extrapolate worldwide ocean Hg(0) emissions. First and foremost, I like the authors' insightful analysis of the observational data and creative approach to estimating oceanic Hg(0) emissions. These contributions are extremely hard and will undoubtedly inspire future research aimed at better understanding the global oceanic Hg cycle using observational data.

We would like to thank the reviewer for taking the time to review our manuscript and for their constructive feedback. Below is our response to the reviewer's comments, organised as follows: the original comments are in blue, our responses are in black, and changes to the manuscript are in red. All line numbers (outside the reviewer's original comments) correspond to the line numbers in the revised manuscript.

While reading the manuscript, I have concerns regarding the accuracy of the authors' methodology for estimating oceanic Hg(0) emissions, or rather, the authors did not adequately explain their method.

- The estimate of dissolved Hg(0) concentrations in surface seawater using steady state air Hg(0) concentrations has significant uncertainty. In this analysis, the authors postulate that steady-state air Hg(0) concentrations result from a balance between gross oceanic Hg(0) evasion and gross air Hg(0) invasion into saltwater. However, considering a box model for Hg(0) cycling, the steady state air Hg(0) concentrations should be also related to atmospheric Hg redox reactions and vertical exchanges between the MBL and free troposphere. The authors did mention atmospheric Hg redox in their discussion and suggested a minimal significance.. I recommend that they calculate the net Hg(0) oxidation fluxes over oceans using model simulations and incorporate these into their equations. Additionally, For the vertical exchange between the MBL and free troposphere, a rough estimate from global models should be beneficial. Hence, I suggest a combination of a global model should be very important for a more accurate estimate.

**1.1. Redox chemistry**

Thank you for the suggestions on how we can improve our model. Following the reviewer's recommendation, we added a net chemical oxidation loss term to the simplified Lagrangian model used to estimate both seawater $Hg^0$ concentrations as well as air-sea $Hg^0$ exchange fluxes, using recent chemical transport model (CTM) simulations from Shah et al. (2021) (see revisions to Appendix B). We found that the derived rate of chemical loss due to oxidation ($R_{ch}$, $2.5 \times 10^{-4}\,h^{-1}$) is two orders of magnitude lower than the rate of $Hg^0_{air}$ concentration change observed at our sites ($A$, $4$–$8 \times 10^{-2}\,h^{-1}$). Recalculating the $Hg^0$ air-sea exchange fluxes incorporating the loss term, we find that it does not have a major impact on the general pattern of flux change with time air masses recently spent in the MBL (Figure 1.1), although divergence between the two estimates

increases with MBL residence time. The values are slightly higher with the incorporation of the loss term, but with an overall increase of only around 0.2 ng m$^{-2}$ h$^{-1}$, and the mean value per site remaining rather small.

[Figure]

**Figure R1.1. Similar to Figure B1 in the manuscript but with some modifications, for the purpose of illustrating incorporating the net oxidation chemical loss term for our Hg$^0$ evasion fluxes. The right and middle panels are exactly the same as those in Figure B1 in the manuscript. On the left panels is the comparison of the flux estimates with and without the incorporation of the chemical loss term. The black line and μ$_F$ value are for the estimates without the chemical loss term, while the red are for the estimates incorporating the chemical loss term.[1]**

For the surface seawater Hg$^0$ concentrations, the new the derivation (see Appendix B):

$$c_w = \frac{c}{H'}\left(1 + \frac{R_{ch}}{A - R_{ch}}\right),$$
* * *
[1] Note that for the "without chemical loss term" experiment (in black, right panels), the curves (i.e., values) are slightly different to those in the original manuscript, as we noticed and have corrected an error in the way we were deriving the flux associated with each trajectory.

has no significant impact on the previous calculations, as the term $\left(1 + \frac{R_{ch}}{A - R_{ch}}\right) \approx 1$, for all sites.

**1.2. Vertical exchange between the MBL and LFT**

We acknowledge the reviewer's comment on the importance of vertical exchange between the MBL and the LFT. However, we believe that adding (another) global CTM estimate(s) to our methodology would increase rather than to decrease the uncertainty of our results. A main factor being the large uncertainty of the bromine oxidant fields used in these models. Adding these CTM-based estimates adds an error that we cannot appropriately quantify.

A central concept/strength of our study was to provide $Hg^0$ air-sea exchange estimates that are independent of previous (CTM-based) estimates. Except for the Lagrangian trajectories, our approach is solely based on a large observational atmospheric $Hg^0$ dataset. These trajectories show that the air masses that arrive at our sampling sites spent the majority of their time in the MBL (see Figure R.2.1 in response to reviewer #2). As such, we argue that vertical exchange between the MBL and LFT would have an insignificant impact on our findings.

We have updated our flux values in the manuscript with new estimates which incorporate the net chemical loss term.

- The calculation of dissolved Hg(0) concentrations in surface seawater based on the steady-state air Hg(0) concentrations indicates a near-zero net ocean Hg(0) emission; that is the gross oceanic Hg(0) emission is balanced by gross air Hg(0) invasion to seawater. Therefore, any further calculations based on the authors' predicted dissolved Hg(0) concentrations in surface seawater will generate zero net ocean Hg(0) emissions. I read from Figure B1 (c, f, I, and I) that positive ocean-air Hg(0) exchange fluxes can be detected in their model, but these net emissions only occurred within a short distance from the site and most of the rest open oceans may not release Hg(0) (net emission) to the air, and this might be inconsistent with traditional knowledge.

First, we would like to clarify a misunderstanding regarding Figure B1. The panels B1c, f, i and I show the $Hg^0$ air-sea exchange flux as function of time air masses recently spent in the MBL, not as a function of distance from the sampling site. What our analyses show is that initially (that is, at the start of the tracking period), there is net $Hg^0$ flux (evasion) from the surface ocean into the air masses, leading to an increase in their atmospheric $Hg^0$ ($Hg^0_{air}$) concentration. This increase is not indefinite. As the air masses travel in the MBL, at a certain point, their $Hg^0_{air}$ concentration reaches equilibrium with the surface ocean dissolved $Hg^0$ ($Hg^0_{aq}$) concentration (i.e., steady state; Figures B1(a, d, g, j)), and net exchange nears zero. This behaviour is consistent with model that describes $Hg^0$ air-sea exchange, $F = k_w(Hg^o{}_{aq} - \frac{Hg^0{}_{air}}{H'})$.

Second, in the applied Lagrangian approach, the steady-state conditions, characterized by nearly constant $Hg^0$ concentrations and a near-zero net air-sea exchange flux, are attributed only to the

air masses that remain in the MBL long enough to reach saturation in their $Hg^0$ concentration. These conditions depend not on the distance from measurement sites but on the time spent in the MBL (this can be seen in Figure 4 in the Manuscript). This assumption does not imply a zero net air-sea exchange flux for the ocean as a whole, since unsaturated air masses continuously enter the MBL from the free troposphere and take up $Hg^0$ from the ocean surface. Thus, our prediction does not suggest zero net ocean $Hg^0$ emissions, as shown in Tables 2 and 3 and Figures B1c, f, I and I of the Manuscript. We have now also clarified in Lines 423 – 427 of the revised Manuscript, that $F \approx 0$ applies only to air masses that have reached steady-state conditions with the underlying surface ocean and that this assumption does not imply net-zero air-sea exchange flux for the whole ocean.

Other minor specific comments:

Line 182-187: better to show the reactive gaseous Hg(II) concentrations and calculate their fractions in total gaseous mercury concentration.

Thank you for the suggestion. Unfortunately, speciated mercury measurements are only available for Amsterdam Island and not for the other sites. At Amsterdam Island, previous work (Angot et al., 2014) has shown that the mean concentration of reactive gaseous mercury ($\sim$ 0.34 pg m$^{-3}$) is more than two orders of magnitude lower than that of $Hg^0$ ($\sim$ 1 ng m$^{-3}$). Given this large difference, we believe that calculating the fraction of reactive gaseous mercury to total gaseous mercury would not provide additional meaningful insights.

Figure 1 c and e, the span of Y axis is to large, making it difficult to read the data.

We agree that the span of the y-axis in Figure 1 is rather large for the Cape Point and Amsterdam Island datasets. The matching y-axis scales were chosen for comparability across the four sites and between the two hemispheres. For better comprehension of the data, we (i) provided an alternative version, Figure S1 (Supplement), where the y-axis scales are adjusted according to each site's data distribution, and (ii) noted in the caption of Figure 1 (in Lines 199 – 200) where the reader can find this alternative version.

We acknowledge, nevertheless, that the note guiding the reader to the alternative figure may have not been clear, and we thank the reviewer for pointing this out. In response, we propose the following revision to Lines 199 – 200:

> Replaced:
> For a zoomed-in view of the data, see Fig. S1 in the Supplement.

> With:
> For better readability of the data, where the y-axis scales are adjusted to each site's data distribution, see Fig. S1 in the Supplement.

Line 307-311: what we can learn from these temporal patterns? Are they indicating ocean Hg(0) emission paly a minor in atmospheric Hg(0)?

The weak seasonal and diurnal variability do not necessarily indicate that ocean emissions play a minor role in atmospheric $Hg^0$ concentrations. On the contrary, our findings in Section 3.3, which show a gradual increase in mean $Hg^0_{air}$ concentration with air mass recent MBL residence time, indicate a significant influence from ocean emissions. Instead, the weak variability may suggest that ocean emissions themselves exhibit relatively weak seasonal and diurnal variability. While we cannot fully discern the contributing factors, one possible explanation is the role of dark reduction in regulating surface ocean $Hg^0_{aq}$ concentrations, as already described in Lines 351 – 358. Lamborg et al. (2021) have proposed that dark reduction, rather than photoreduction, is the dominant source of $Hg^0_{aq}$ in the surface ocean. Since dark reduction is largely independent of solar variability, it could maintain relatively stable (that is, exhibiting weak seasonal and diurnal variability) surface ocean $Hg^0_{aq}$ concentrations, leading to relatively stable emissions and consequently weak seasonal and diurnal variability in MBL $Hg^0_{air}$ concentrations.

Line 364-365: did the authors calculate the mean air Hg(0) concentrations for the LFT categories? What is the difference between them and those in the MBL? Are they similar to those expected here?

We did not calculate mean $Hg^0_{air}$ concentrations for the LFT categories in this study. Instead, our hypothesis is based on the difference in mean $Hg^0_{air}$ concentrations between the MBL (from this study) and the LFT (from previous studies), as outlined in Lines 341 – 349. While LFT $Hg^0_{air}$ concentrations from ground-based monitoring sites can be possible under certain conditions, for example on high-altitude mountain observatories that receive a lot of direct LFT influence (e.g., Koenig et al., 2023), our sites primarily sample MBL air, making direct LFT estimates from the data challenging.

Line 394-395: given the air mass travelling time and atmospheric Hg(0) residence time, about 3% loss of air Hg(0) can be expected during 6 days due to oxidation. While from Figure 4, the mean air Hg(0) concentrations during the initial several days are generally less than 10% lower than the steady state concentrations.

As discussed above, following the reviewer's suggestion, we included the effect of redox chemistry in our Lagrangian model used to derive the $Hg^0$ air-sea exchange fluxes. With its inclusion, we showed that the rate of chemical loss due to oxidation is two orders of magnitude lower than the rate of $Hg^0$ concentration change (which we assume is driven primarily by $Hg^0$ air-sea exchange). Hence, the observed $Hg^0_{air}$ concentration increase far exceeds the loss from oxidation, explaining the observed increase rather than an expected oxidation-driven decrease over the 6 days.

Of course, from Figure R1.1, we see that the divergence between the instantaneous flux estimates derived with and without inclusion of the chemical loss term increases with air mass recent MBL residence time. This suggests that net oxidation becomes increasingly relevant over

longer time scales. However, over the relatively short time scale considered here, the effect of net oxidation is negligible.

We have also included this new information in the paragraph referring to the potential influence of redox chemistry on the patterns observed in Figure 4, Lines 391 – 399, as follows:

> Replaced:
> Nevertheless, the observed asymptotic Hg0air increase at the sites may be partially explained by the exchange of $Hg^0$ between the ocean and the atmosphere. The pattern is consistent with the air–sea gas exchange model (Eq. (1) in Sect. 3.5), which predicts net ocean $Hg^0$ emissions to the surrounding air until the $Hg^0_{air}$ concentration normalised by the Henry's law constant matches the surface ocean $Hg^0_{aq}$ concentration. While other processes may be involved, we are unable to discern them in this study. Given the short travel time considered here (6 days) relative to $Hg^0$'s average atmospheric lifetime against oxidation (6 months–1 year; Horowitz et al., 2017; Saiz-Lopez et al., 2018; Shah et al., 2021), oxidation of $Hg^0_{air}$ is likely insignificant.
>
> With:
> Nevertheless, the observed asymptotic Hg0air increase at the sites may be partially explained by the exchange of Hg0 between the ocean and the atmosphere. The pattern is consistent with the air–sea gas exchange model (Eq. (1) in Sect. 3.5), which predicts net ocean $Hg^0$ emissions into passing air masses until the Hg0air concentration in them normalised by the Henry's law constant matches the surface ocean $Hg^0_{aq}$ concentration. While other processes may be involved, we are unable to discern them in this study. Concerning atmospheric redox chemistry, our estimation of the rate of net chemical loss due to oxidation shows that it is two orders of magnitude lower than the rate of $Hg^0$ concentration change observed here (see details in Appendix B), suggesting that the observed $Hg^0_{air}$ concentration increase far exceeds net oxidation-driven loss. Over longer periods, net oxidation may play a more significant role in shaping Hg0 air concentrations in the MBL. However, over the short time scale considered here, its impact on the observed patterns is negligible.

Line 433-435: with these dissolved Hg(0) concentrations in surface seawater, we will expect zero net ocean Hg(0) emissions according the equation (1).

As addressed in a previous response, in the Lagrangian approach applied in our study, steady-state conditions, characterized by nearly constant $Hg^0_{air}$ concentrations and a near-zero net air-sea exchange flux, are attributed only to the air parcels that remain in the MBL long enough to reach saturation in $Hg^0_{air}$ concentration. This assumption does not imply a zero net air-sea exchange flux for the ocean as a whole, since unsaturated air masses continuously enter the MBL from the free troposphere and take up $Hg^0$ from the ocean surface. Thus, our prediction does not suggest zero net ocean $Hg^0$ emissions.

Line 455-457: Previous studies may have reported numerous dissolved Hg(0) concentrations in surface seawater, which should encompass DGM concentrations during both enriched and un-enriched DGM episodes. Therefore, short-term measurements of DGM might be not the reason causing the elevated DGM concentrations by previous studies as compared with this study.

We agree with the reviewer that a combined dataset from multiple short-term measurements can capture the full variability in DGM concentrations, including both enriched and unenriched episodes. In Lines 455–457, we are specifically referring to the comparison of our estimates, which are based on multi-year ($Hg^0_{air}$) observations, to individual measurement campaigns, which typically span a few days to a few weeks. We acknowledge that this may have not been articulated clearly. The point we tried to emphasise was that short-term measurements likely capture transient conditions, whereas our estimates represent long-term averages. We thank the reviewer for pointing this out, and propose the following revision to Lines 463 – 468:

Replaced:
In particular, the concentrations reported in the literature are based on measurements performed over very short periods (days to a few weeks; see Tables 2 and 3, for example), whereas our estimates are derived from ($Hg^0_{air}$) observations spanning multiple years.

With:
While the combination of measurements from multiple short-term studies can capture the full variability in surface ocean $Hg^0_{aq}$ concentrations, individual measurement campaigns, which typically span a few days to a few weeks (see Tables 2 and 3, for example), primarily reflect transient conditions. In contrast, our estimates, which are derived from multi-year ($Hg^0_{air}$) observations, represent long-term averages.

In conclusion, the manuscript is well-organized and well-written. The concept of using observational data to constrain oceanic Hg(0) emissions is innovative. However, it remains unclear whether the proposed method for estimating ocean Hg(0) emissions is feasible. This warrants further evaluation by the authors.

Thank you once again for the kind words and insightful feedback. The reviewer's comments have helped improve our manuscript by highlighting aspects which we had overlooked. Our assumption that the atmospheric $Hg^0$ oxidation is insignificant has been proven wrong and we amended our method to include its effect on our estimates. We hope we have fully and clearly addressed reviewer's concerns.

**References**

Angot, H., Barret, M., Magand, O., Ramonet, M., and Dommergue, A.: A 2-year record of atmospheric mercury species at a background Southern Hemisphere station on Amsterdam Island, Atmos. Chem. Phys., 14, 11461–11473, https://doi.org/10.5194/acp-14-11461-2014, 2014.

Koenig, A. M., Magand, O., Verreyken, B., Brioude, J., Amelynck, C., Schoon, N., Colomb, A., Ferreira Araujo, B., Ramonet, M., Sha, M.K., and Cammas, J. P.: Mercury in the free troposphere and bidirectional atmosphere–vegetation exchanges–insights from Maïdo mountain observatory in the Southern Hemisphere tropics, Atmos. Chem. Phys., 23, 1309–1328, https://doi.org/10.5194/acp-23-1309-2023, 2023.

Lamborg, C. H., Hansel, C. M., Bowman, K. L., Voelker, B. M., Marsico, R. M., Oldham, V. E., Swarr, G. J., Zhang, T., and Ganguli, P. M.: Dark reduction drives evasion of mercury from the ocean, Front. Environ. Chem., 2, 659085, https://doi.org/10.3389/fenvc.2021.659085, 2021.

Shah, V., Jacob, D. J., Thackray, C. P., Wang, X., Sunderland, E. M., Dibble, T. S., Saiz-Lopez, A., Černušák, I., Kellö, V., Castro, P. J., Wu, R., and Wang, C.: Improved Mechanistic Model of the Atmospheric Redox Chemistry of Mercury, Environ. Sci. Technol., acs.est.1c03160, https://doi.org/10.1021/acs.est.1c03160, 2021.